# Transformers Can Learn Posterior Predictive Distributions In-Context

Gyeonghun Kang [1]  Changwoo J. Lee [1] [†]  Xiang Cheng [2] [†]

## Abstract

Prior-data fitted networks (PFNs) have recently emerged as a powerful approach for Bayesian prediction tasks, approximating the posterior predictive distribution (PPD) through in-context learning. Despite their strong empirical performance and ability to go beyond point predictions, theoretical understandings of the algorithmic capability of transformers to learn distributions in context are still lacking. Focusing on Gaussian process regression problems, we show by construction that transformers can implement a gradient descent algorithm targeting the posterior predictive mean and variance, followed by nonlinear mappings that yield binned probabilities of PPD. We study the error bounds of the approximated PPD in terms of attention depth and bin resolution. Based on these results, we further demonstrate the key role of normalization and the choice of attention depth in enabling the extrapolation abilities of transformers beyond the pretraining sample size range. We conduct simulations that corroborate our findings, providing insight into the expressivity of PFNs targeting PPDs and how architectural choices may influence generalization capabilities.

## 1. Introduction

In-context learning (ICL) is the ability of a pre-trained model to adapt to new tasks at inference time by using examples provided in the input sequence without any parameter updates (Brown et al., 2020; Garg et al., 2022). This ability has been most prominently observed in transformer architectures, which have become the dominant modeling component in modern machine learning (Vaswani et al., 2017). Formally, we refer to ICL as the ability to learn a mapping from a set of input-output pairs $\mathcal{D}_n = \{(x_i, y_i)\}_{i=1}^n$

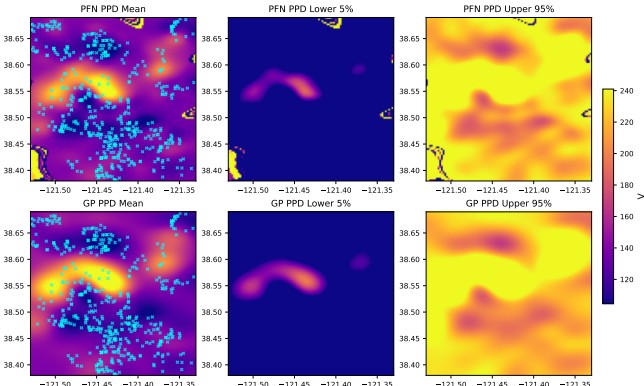

*Figure 1.* Comparison of PPDs produced by a transformer-based PFN and an empirical-Bayes Gaussian process (GP) on the Sacramento home price dataset (Kuhn, 2008). The axes represent spatial coordinates (longitude and latitude), and the color scale represents price per square foot (V). The top row shows PFN posterior predictive summaries, and the bottom row shows the corresponding GP outputs. Columns correspond to the PPD mean (left), $5\%$ quantile (center), and $95\%$ quantile (right). Housing locations used as context are shown by blue $\times$ marks in the left column. The PFN PPD tracks that of exact GP closely overall except in the regions with no nearby observations. See Section F for details.

together with a new input $x_{n+1}$ to a prediction of $y_{n+1}$; see Dong et al. (2024) and references therein for a review.

When the goal is probabilistic prediction of $y_{n+1}$, rather than point prediction, prior-data fitted networks (PFNs) (Müller et al., 2022) have recently gained considerable attention as an ICL method for approximating the posterior predictive distribution (PPD) of $y_{n+1}$. By pre-training a transformer model on synthetic data from a joint prior predictive model of $(x, y)$, PFNs aim to learn a mapping from $(\mathcal{D}_n, x_{n+1})$ to the posterior predictive distribution $p(\cdot|x_{n+1}, \mathcal{D}_n)$. Compared to traditional approaches for computing PPDs such as Markov chain Monte Carlo (MCMC), PFNs offer substantial computational gains since model parameters are no longer updated after pre-training and also maintain robustness through pre-training on diverse data-generating scenarios. Most notably, PFNs for tabular data have demonstrated strong empirical performance and excellent scalability across a wide range of benchmarks (Hollmann et al., 2023; 2025). PFNs have also been adopted in a broad range of application domains, such as RNA biophysics (Scheuer et al., 2025) and genomic prediction

---
[†]Equal advising. [1]Department of Statistical Science, Duke University, Durham, NC, USA [2]Department of Electrical and Computer Engineering, Duke University, Durham, NC, USA. Correspondence to: Gyeonghun Kang <gyeonghun.kang@duke.edu>.

*Proceedings of the 43rd International Conference on Machine Learning*, Seoul, South Korea. PMLR 306, 2026. Copyright 2026 by the author(s).

(Ubbens et al., 2025).

We are particularly motivated by PFN's ability to quantify posterior predictive uncertainty. For instance, under Gaussian process regression settings, Müller et al. (2022) reports that 95% credible intervals of $y_{n+1}$ produced by PFNs are virtually indistinguishable from the truth; see also Zhang et al. (2025), where PFN-based PPD intervals closely match nominal coverage under interpolation settings. Also, PFNs have been shown to successfully serve as flexible surrogate models for Bayesian optimization (Müller et al., 2023; Yu et al., 2026), accurately computing common acquisition functions such as expected improvement, which depend on both posterior predictive means and variances. See Figure 1 for an illustration of PFN's ability to accurately capture the lower 5% and upper 95% PPD quantiles; details are in Section F.

Despite its increasing popularity, the theoretical understanding of PFNs remains largely unexplored, especially regarding the role of the transformer in the pre-training stage. An increasing body of work suggests that ICL can be interpreted as implicitly performing gradient-based optimization. For a linear regression task, Akyürek et al. (2023); Von Oswald et al. (2023); Bai et al. (2023) demonstrates that transformers and their recurrences are expressive enough to implement gradient descent updates in their forward pass; see also Fu et al. (2024); Vladymyrov et al. (2024). Furthermore, Ahn et al. (2023); Zhang et al. (2024); Mahankali et al. (2024) shows that the global optimum of a single layer transformer weights in the linear regression task implements a single iteration of preconditioned gradient descent. For PFNs, a natural research question concerns understanding how PFNs carry out in-context learning of PPD, and the mechanisms and conditions under which this capability arises.

Taking a gradient-based optimization view of ICL, we are particularly interested in a principled understanding of how attention depth affects the approximation quality of PPDs. Recent state-of-the-art PFN models suggest that increasing network depth can substantially improve performance. For example, TabPFN-2.5 and TabPFN-3 (Grinsztajn et al., 2025; 2026) report markedly improved accuracy by increasing attention depth to 18 or 24 layers, a 1.5- or 2-fold increase over TabPFNv2 (Hollmann et al., 2025) with 12 layers. Another important aspect of PFN that affects the approximation quality of PPDs is the discretization resolution used for PPDs over continuous domains, a practice that is widespread in existing PFN implementations (Müller et al., 2022; Hollmann et al., 2025) but is seldom studied in the literature.

Another interesting aspect of PFNs is their ability to generalize beyond the pre-training sample size. For instance, even though TabPFNv2 is pretrained on synthetic data only up to a sample size of 2048, it remains highly competitive

for context size $n$ up to 10,000 (Hollmann et al., 2025). To reach even larger, recent works have adopted approaches such as fine-tuning (Feuer et al., 2024), boosting (Wang et al., 2025), and two-stage architectures (Qu et al., 2025). Adopting a gradient-based optimization view of ICL, we seek to provide a better understanding of when this generalization ability arises and how it can be further strengthened. Nagler (2023) attempts to explain this capability asymptotically, based on sensitivity to individual training samples and localization for a fixed architecture; our focus is instead on the relationship between attention depth, normalization, and generalization ability.

**Contributions.** Focusing on GP regression problems, we establish theoretical support for PFNs' capability to approximate the PPD and identify the mechanisms driving the generalization across different context size $n$, referred to as $n$-*generalization*, whereby models retain reliable point predictions and uncertainty quantification beyond the pretraining sample size range.

1. We provide an explicit transformer construction that approximates the PPD in-context up to a prescribed tolerance: self-attention implements an iterative solver for the predictive mean and variance, and a shallow multilayer perceptron (MLP) maps these moments to a binned distribution. With binning and tail errors controlled, the principal remaining error arises in the attention block (Theorem 4.1), which inherits an exponential convergence rate in the number of layers $L$ (Theorem 3.1). This extends earlier ICL works focused on point estimation to distributional approximation assessment.

2. We use our construction to characterize both the mechanisms that enable and the factors that limit $n$-generalization in PFNs under a controlled setting. Using the spectral properties of the attention score matrix, we show that attention normalization is essential for achieving generalization (Theorem 5.1, Figure 4). We then demonstrate that extending generalization to substantially larger $n$ is intrinsically more difficult (Theorem 5.2), and explain how the increased attention depth could help mitigate this challenge (Theorem 5.3).

3. Using our transformer pretrained in PFN fashion, we empirically validate our theory: PPD approximation error decreases with attention depth and bin resolution, in line with Theorem 4.1 (Figure 2). We further show that attention normalization is crucial for $n$-generalization: unnormalized models fail outside the pretraining range, while normalized models remain stable (Figure 4). Finally, we confirm that generalization performance improves with both depth and pretraining range, and that solver errors grow with evaluation sample size, consistent with Theorem 5.3 (Figure 5, 10).

## 2. Background and Problem Setup

**PFNs and in-context learning of PPD.** We begin by introducing the necessary background and notation for in-context learning of posterior predictive distributions (PPD) and prior-data fitted networks (PFN). Let $p(y \mid x, \theta)$ denote a conditional probability model of $y \in \mathcal{Y} \subseteq \mathbb{R}$ parametrized by $\theta$ with a prior $p(\theta)$, and $x \in \mathbb{R}^d$ is a covariate. Given i.i.d. data $\mathcal{D}_n = \{(x_i, y_i)\}_{i=1}^n$, the PPD of $y$ at a query $x$ is

$$p_n(y \mid x, \mathcal{D}_n) = \int p(y \mid x, \theta) p(\theta \mid \mathcal{D}_n) d\theta \qquad (1)$$

where $p(\theta \mid \mathcal{D}_n)$ is a posterior of $\theta$. The integral in (1) is often analytically intractable, and PPD is typically computed through Monte Carlo integration using samples from the posterior $p(\theta \mid \mathcal{D}_n)$, such as those from MCMC.

Without appealing to expression (1), prior-data fitted networks (PFNs) (Müller et al., 2022) take a radically different approach by directly approximating PPD through in-context learning. Assuming a joint prior predictive model $(x_i, y_i) \overset{\text{iid}}{\sim} p(x, y)$ for all $i$, PFN aims to learn a mapping $(\mathcal{D}_n, x) \mapsto q_\vartheta(\cdot \mid x, \mathcal{D}_n)$ that accepts a dataset $\mathcal{D}_n$ and a query $x$ as input and outputs a probability distribution $q_\vartheta$ that directly approximates PPD (1). The mapping, parameterized by $\vartheta$, corresponds to a forward pass of a neural net, and $\vartheta$ is optimized such that

$$\vartheta = \operatorname{argmin} \mathrm{E}_{\mathcal{D}_n \cup \{x, y\}} [-\log q_\vartheta(y \mid x, \mathcal{D}_n)]$$

where expectation is taken over the assumed joint prior predictive model $(x_i, y_i) \overset{\text{iid}}{\sim} p(x, y)$. This is equivalent to minimizing the expected KL divergence between PPD and $q_\vartheta$; see Müller et al. (2022) for details. In practice, $\vartheta$ is optimized such that the resulting mapping can approximate PPD for a broad range of context sizes $n$. To achieve this, PFN is pre-trained on ensembles of synthetic data with varying sample sizes; see also Nagler (2023).

**PFN outputs and population loss minimizer.** We focus on PFN based on transformer consisting of $L$ self-attention blocks and an MLP head with parameter $\vartheta$ that yields a vector of finite length $C$ ("logits"), denoted as $\ell = (\ell_1, \ell_2, \dots, \ell_C)$, where its softmax $\operatorname{sm}(\ell)_c := \exp(\ell_c) / \sum_{c'=1}^C \exp(\ell_{c'})$ and $c = 1, \dots, C$ represent the PPD output $q_\vartheta$. Specifically, we fix a partition $\Gamma = \{a = \gamma_1 < \gamma_2 < \cdots < \gamma_{C+1} = b\}$ of an interval $(a, b) \subset \mathcal{Y}$, such that marginally $\mathbb{P}(y \notin (a, b]) = \varepsilon$ for a small $\varepsilon > 0$. Then, the logits $(\ell_1, \dots, \ell_C)$ induce a piecewise-constant probability density function of $y$ for $y \in (a, b]$:

$$q_\vartheta(y \mid x, \mathcal{D}_n) := \sum_{c=1}^C \frac{\mathbf{1}_{y \in (\gamma_c, \gamma_{c+1}]}}{\Delta_c} \frac{\exp(\ell_c)}{\sum_{c'=1}^C \exp(\ell_{c'})} \qquad (2)$$

where $\Delta_c := \gamma_{c+1} - \gamma_c$. This setting coincides with a state-of-the-art practical implementation of PFNs that yields

a piece-wise constant output distribution of $y$ (Hollmann et al., 2025).

The network parameter $\vartheta$, whose architecture and details will be described in the following section, is pre-trained using simulated datasets from the assumed joint prior predictive model $(x, y) \overset{\text{iid}}{\sim} p(x, y)$. It minimizes the (truncated) negative log likelihood (NLL) loss $\mathrm{E}_{\mathcal{D}_n \cup \{x\}}[\mathrm{E}_{y \sim p_{(a,b]}(\cdot \mid x, \mathcal{D}_n)} \{-\log q_\vartheta(y \mid x, \mathcal{D}_n) \mid x, \mathcal{D}_n\}]$, and the population minimizer $q^*$ is the truncated and discretized PPD (see Lemma B.7):

$$q^*(y \mid x, \mathcal{D}_n) := \sum_{c=1}^C \frac{\mathbf{1}_{y \in (\gamma_c, \gamma_{c+1}]} \int_{\gamma_c}^{\gamma_{c+1}} p_n(t \mid x, \mathcal{D}_n) dt}{\Delta_c (1 - \varepsilon)}. \quad (3)$$

Assuming $p_n$ is Lipschitz smooth on $(a, b)$, one can show that $\mathrm{E}_{\mathcal{D}_n \cup \{x\}}[\mathrm{TV}(p_n, q^*)] = O(|\Gamma|) + \varepsilon$, where TV is the total variation distance and $|\Gamma|$ is the maximum bin width of $\Gamma$; see Lemma B.1 for details.

**Transformer architecture.** We consider a transformer architecture consisting of $L$ self-attention blocks followed by an MLP head. Let $Z^{(0)} \in \mathbb{R}^{(d+1) \times (n+1)}$ be an input token matrix where each column is a token:

$$[Z^{(0)}]_{:,j} = [x_j^\top, y_j]^\top \ (j \le n), \quad [Z^{(0)}]_{:,n+1} = [x_{n+1}^\top, 0]^\top,$$

so the $(n + 1)$st token corresponds to the query point with a masked label. Denoting $\operatorname{Attn}$ a masked self-attention (see Section 3), we define an update for $Z^{(l+1)} \in \mathbb{R}^{(d+1) \times (n+1)}, l = 0, \dots, L-1$ and the output logits $\ell$ as

$$\begin{aligned} Z^{(l+1)} = \ & Z^{(l)} + S^{(l)} Z^{(l)} \\ & + \operatorname{Attn}(Z^{(l)}; V^{(l)}, K^{(l)}, Q^{(l)}), \end{aligned} \qquad (4)$$

$$\ell_\vartheta(Z^{(0)}) := W_2 \operatorname{act}(W_1 [Z^{(L)}]_{:,n+1} + h_1) + h_2, \quad (5)$$

where $V^{(l)}, K^{(l)}, Q^{(l)}, S^{(l)} \in \mathbb{R}^{(d+1) \times (d+1)}$ are the per-layer parameters (attention projections and linear skip connections); $W_1 \in \mathbb{R}^{C' \times (d+1)}, W_2 \in \mathbb{R}^{C \times C'}, h_1 \in \mathbb{R}^{C'}$, and $h_2 \in \mathbb{R}^C$ are the MLP-head parameters, and $\operatorname{act}(\cdot)$ is an activation function. The update (4) comprises a residual connection, a self-attention map, and an additional learnable skip term. We write $\vartheta$ for the collection of all parameters. This abstraction is sufficient for our constructive proofs; additional components used in practice (e.g., token embeddings, multi-head attention, and layer normalization) can be viewed as architectural refinements that increase expressivity without altering the core mechanism.

It has been shown that the attention blocks (4) are expressive enough to implement an iterative solver computing the predictive mean of $p_n$ (Von Oswald et al., 2023; Ahn et al., 2023; Cheng et al., 2024). However, approximating an entire predictive distribution is more demanding: it requires matching not only the mean but also higher-order characteristics, such as variance. In the following sections, we address this gap via a constructive proof.

**Gaussian process (GP) regression.** We consider the GP regression problem (Rasmussen & Williams, 2005) as a canonical example since the tractability of the PPD arising from the GP regression problem allows us to carefully examine the accuracy of the PFN output $q_\vartheta$. Let $\phi : \mathbb{R}^d \to \mathbb{R}$ be an unknown function, and $y_1, \ldots, y_n$ be a noisy realization of $\phi$ based on a probabilistic model

$$y_i = \phi(x_i) + \epsilon_i, \quad \epsilon_i \overset{\text{iid}}{\sim} \mathcal{N}(0, \sigma^2). \quad (6)$$

Also, let $Y := (y_1, \ldots, y_n)^\top$ be the training labels, $X := (x_1^\top, \cdots, x_n^\top)^\top$ be the feature matrix, and let $Z := (x, \mathcal{D}_n)$ denote the query-context pair. Assuming fixed error variance $\sigma^2$, along with a mean zero GP prior on $\phi$ with a covariance kernel $\kappa : \mathbb{R}^d \times \mathbb{R}^d \to \mathbb{R}$, the PPD $p_n(y \mid x, \mathcal{D}_n)$ is also a normal distribution $\mathcal{N}(y; \mu(Z), \tau(Z))$ where

$$\mu(Z) = k_x^\top (G + \sigma^2 I_n)^{-1} Y,$$
$$\tau(Z) = \kappa(x, x) + \sigma^2 - k_x^\top (G + \sigma^2 I_n)^{-1} k_x$$

where $k_x := \big(\kappa(x_1, x), \ldots, \kappa(x_n, x)\big)^\top$ and $G \in \mathbb{R}^{n \times n}$ with $[G]_{i,j} = \kappa(x_i, x_j)$ is a Gram matrix. Thus, both $\mu(Z)$ and the variance reduction term $k_x^\top (G + \sigma^2 I_n)^{-1} k_x$ reduce to evaluating $k_x^\top (G + \sigma^2 I_n)^{-1} v$ for $v \in \{Y, k_x\}$.

## 3. Attention Computes Moments of PPD

We exhibit an explicit weight configuration under which attention implements the iterative recursions for the posterior predictive mean $\mu(Z)$ and variance $\tau(Z)$.

**KRR and Richardson iteration.** Our starting point is to view $\mu(Z)$ and the variance reduction term in $\tau(Z)$ as arising from kernel ridge regression (KRR) problems. Let $\mathcal{H}$ be the reproducing kernel Hilbert space (RKHS) of $\kappa$ with norm $\| \cdot \|_\mathcal{H}$. For $v \in \mathbb{R}^n$, define $u^* \in \mathcal{H}$ as

$$u^* = \arg \min_{u \in \mathcal{H}} \sum_{i=1}^n (v_i - u(x_i))^2 + \sigma^2 \|u\|_\mathcal{H}^2.$$

By the representer theorem (Kimeldorf & Wahba, 1970), $u^*(x) = k_x^\top (G + \sigma^2 I_n)^{-1} v$, and therefore $u_X^* := (u^*(x_1), \ldots, u^*(x_n))^\top$ is the solution of $(G + \sigma^2 I_n) u_X^* = Gv$. A Richardson iteration (Richardson, 1911) for this system can be written componentwise as, for $j = 1, \ldots, n$,

$$u^{(l+1)}(x_j) = (1 - \eta^{(l)} \sigma^2) u^{(l)}(x_j)$$
$$+ \eta^{(l)} \sum_{i=1}^n \kappa(x_i, x_j)\big(v_i - u^{(l)}(x_i)\big), \quad (7)$$

initialized at $u^{(0)}(\cdot) \equiv 0$ and extended to a query $x$ by the same update, with $x_j$ replaced by $x$.

Let $\lambda_1(G)$ and $\lambda_n(G)$ denote the maximum and minimum eigenvalues of $G$. If $0 < \eta^{(l)} < 2/(\lambda_1(G) + \sigma^2)$ for

all $l$ and $\sum_{l=0}^\infty \eta^{(l)} = \infty$, then as $L \to \infty$, $u_X^{(l)} := (u^{(l)}(x_1), \cdots, u^{(l)}(x_n))^\top \to u_X^*$, and $u^{(L)}(x) \to k_x^\top (G + \sigma^2 I_n)^{-1} v$ (see Theorem B.8). In particular, $u^{(L)}(x) \to \mu(Z)$ when $v = Y$, and $u^{(L)}(x) \to k_x^\top (G + \sigma^2 I_n)^{-1} k_x$ when $v = k_x$. Therefore, we can compute the moments of PPD by running the recursion (7) for two choices of the righthand side: $v = Y$ and $v = k_x$.

**Unnormalized attention.** These recursions can be realized through attention layers. Let $Z = [z_1, \ldots, z_n, z_{n+1}] \in \mathbb{R}^{d' \times (n+1)}$ be a matrix of tokens, where $d'$ is a token dimension. We define *unnormalized* attention $\text{Attn}_{M, \kappa} : \mathbb{R}^{d' \times (n+1)} \to \mathbb{R}^{d' \times (n+1)}$ as

$$\text{Attn}_{M, \kappa}(Z; V, K, Q) := VZh_{M, \kappa}(KZ, QZ) \quad (8)$$

where $M \in \{0, 1\}^{(n+1) \times (n+1)}$ is a binary mask, and $h_{M, \kappa} : \mathbb{R}^{d' \times (n+1)} \times \mathbb{R}^{d' \times (n+1)} \to \mathbb{R}^{(n+1) \times (n+1)}$ satisfies $[h_{M, \kappa}(KZ, QZ)]_{ij} = \kappa(Kz_i, Qz_j) M_{ij}$ for a positive semidefinite kernel $\kappa : \mathbb{R}^{d'} \times \mathbb{R}^{d'} \to \mathbb{R}$. Equivalently, the $j$th output token is

$$[\text{Attn}_{M, \kappa}(Z; V, K, Q)]_{:,j} = \sum_{i=1}^{n+1} \kappa(Kz_i, Qz_j) M_{ij} V z_i.$$

Examples of kernels include the *linear* kernel $\kappa(x, x') = x^\top x'$ and the *radial basis function (RBF)* kernel $\kappa(x, x') = \exp(-\|x - x'\|^2 / 2)$.

**Attention computes $\mu(Z)$ and $\tau(Z)$ in-context.** The attention block in (4) can be configured to simultaneously carry out both recursions in parallel at each layer. The decay term $(-\eta^{(l)} \sigma^2)$ in (7) is implemented via the additional learnable connection, while the remaining term is realized by self-attention over the context tokens. Consequently, the transformer inherits the same convergence rate as the underlying Richardson iteration, which is exponential in $L$.

**Theorem 3.1** (Attention computes PPD moments). *Define* $\text{TF}_{\vartheta, L}(Z^{(0)}) := W[Z^{(L)}]_{:,n+1}$ *as*

$$Z^{(0)} = \begin{bmatrix} X^\top & x_{n+1} \\ Y^\top & 1 \\ 0_{3 \times n} & 0_3 \end{bmatrix} \in \mathbb{R}^{(d+4) \times (n+1)}$$
$$Z^{(1)} = Z^{(0)} + \text{Attn}_{M^{(0)}, \kappa}(Z^{(0)}; V^{(0)}, K^{(0)}, Q^{(0)}) \quad (9)$$
$$Z^{(l+1)} = Z^{(l)} + \text{Attn}_{M, \kappa}(Z^{(l)}; V^{(l)}, K^{(l)}, Q^{(l)})$$
$$+ S^{(l)} Z^{(l)}, \quad l = 1, \cdots, L - 1, \quad (10)$$

*where* $M_{i,j}^{(0)} = 1_{\{i > n\}}$, $M_{i,j} = 1_{\{i \leq n\}}$ *are attention masks and* $W \in \mathbb{R}^{2 \times (d+4)}$ *is a readout matrix. Then there exists* $\vartheta$, *a set of all parameters (given in* (23)*), such that*

$$\left\| \text{TF}_{\vartheta, L} - (\mu, \tau)^\top \right\|_\infty \lesssim \exp(-(1 - \rho)L).$$

*where* $\rho := 1 - \eta(\lambda_n(G) + \sigma^2) \in (0, 1)$ *is the convergence factor, and* $\eta$ *is the step size encoded in* $V^{(l)}$ *and* $S^{(l)}$.

Thus, $\mathrm{TF}_{\vartheta,L}$ functions as an approximate solver with a finite iteration budget $L$. This observation has direct implications for context size generalization, which we detail in Section 5.

*Remark* 3.2 (Extension to hierarchical GPs). The construction in Theorem 3.1 can be potentially extended to fully Bayesian GP regression; see Section E for related experiments. Specifically, when the resulting PPD is a finite mixture of componentwise GP predictive distributions, one can enlarge the token dimension and use multi-head attention so that each head runs the Richardson-style recursion from Theorem 3.1 for one component. This would compute the componentwise predictive moments, while the mixture weights could, in principle, be represented by an additional readout module. A full treatment of this hierarchical extension is left for future work.

## 4. From Moments to PPD

The MLP head in (5) can map the moments computed by $\mathrm{TF}_{\vartheta,L}$ to a density. In particular, a shallow MLP followed by a softmax is naturally suited to density approximation via discretization. To make this concrete, consider a one-dimensional exponential family density $f_\theta$ on $\mathcal{Y} \subseteq \mathbb{R}$ with parameter $\theta \in \Theta \subseteq \mathbb{R}^p$:

$$f_\theta(y) = \exp\left(\langle \psi(\theta), T(y) \rangle - A(\theta)\right) h(y),$$

where $\psi : \Theta \to \mathbb{R}^{p'}$ is the natural parameter map, $T : \mathcal{Y} \to \mathbb{R}^{p'}$ is the sufficient statistic map, $A(\theta)$ is the log normalizing constant, and $h$ is the base density. For example, for a Gaussian $\mathcal{N}(\mu, \tau)$, one may take $\psi(\mu, \tau) = \left(\mu/\tau, -1/(2\tau)\right)$ and $T(y) = (y, y^2)$. We further make $h$ a constant by expanding $\psi$ and $T$ to absorb the non-constant term in $h$ if any; see Section C.2 for details.

**MLP for natural parameter conversion.** Let $\mathcal{K} \subset \Theta$ be compact. By the universal approximation theorem (Hornik, 1991; Leshno et al., 1993), for any $\delta_{\mathrm{MLP}} > 0$, there exists a one-hidden-layer MLP $\tilde{\psi} : \mathcal{K} \to \mathbb{R}^{p'}$ of the form

$$\tilde{\psi}(\theta) = \tilde{W}_2 \, \mathrm{act}(\tilde{W}_1 \theta + \tilde{h}_1) + \tilde{h}_2,$$

with width $C'$ and a non-polynomial activation $\mathrm{act}$ (e.g., ReLU), such that $\sup_{\theta \in \mathcal{K}} \|\psi(\theta) - \tilde{\psi}(\theta)\|_2 \le \delta_{\mathrm{MLP}}$.

**Softmax discretization.** Let $\Gamma = \{\gamma_c\}_{c=1}^{C+1}$ be an equidistant partition of $(a, b) \subset \mathcal{Y}$, with midpoints $\xi_c := (\gamma_c + \gamma_{c+1})/2$. Define a readout matrix $\Xi \in \mathbb{R}^{C \times p'}$ by $\Xi_{c,:} := T(\xi_c)^\top$. Then $\Xi\tilde{\psi}(\theta)$ is still an MLP output, corresponding to $W_2 = \Xi\tilde{W}_2, W_1 = \tilde{W}_1 W, h_1 = \tilde{h}_1, h_2 = \Xi\tilde{h}_2$ in expression (5) for GP regression, and $(\Xi\tilde{\psi}(\theta))_c = \langle \tilde{\psi}(\theta), T(\xi_c) \rangle$ becomes the inner product between the natural parameter and the sufficient statistics evaluated at $\xi_c$. Moreover, softmax normalization removes the additive constant $A(\theta)$ in the logits. Thus, with a constant $h$, the piecewise-constant

density

$$\tilde{f}_\theta(y) := \sum_{c=1}^{C} \frac{\mathbf{1}_{y \in (\gamma_c, \gamma_{c+1}]}}{(b-a)/C} \, \mathrm{sm}\left(\Xi\tilde{\psi}(\theta)\right)_c$$

constitutes a midpoint discretization that can approximate $f_\theta$ arbitrarily well on $(a, b)$ as $C \to \infty$ (up to the MLP error; see Lemma B.2).

Putting the pieces together, deeper attention depths ($L$) control the convergence of the moments computation recursion, while larger $C$ controls the discretization error. The following theorem summarizes the resulting approximation guaranty (up to truncation mass outside $(a, b)$); see Figure 2.

**Theorem 4.1** (Transformer PPD approximation). *Under the regularity conditions stated in Section C.2, there exists $\vartheta$ such that, for any context $Z^{(0)}$, the total variation distance between the true posterior predictive $p_n(\cdot \mid Z^{(0)})$ and the Transformer output $q_\vartheta(\cdot \mid Z^{(0)})$ satisfies*

$$\mathrm{TV}\left(p_n, q_\vartheta\right) \lesssim e^{-(1-\rho)L} + \delta_{\mathrm{MLP}} + \frac{1}{C} + \varepsilon_{tail}(Z^{(0)}),$$

*where $\rho \in (0, 1)$ is the convergence factor of the attention block and $\varepsilon_{tail}(Z^{(0)}) := \mathbb{P}_{y \sim p_n(\cdot \mid Z^{(0)})}\left(y \notin (a, b)\right)$.*

Theorem 4.1 makes explicit how architectural choices relate to approximation accuracy. In practice, the attention stack is often the main architectural component of interest, whereas the truncation interval, $C$, and the MLP width are typically chosen large enough (Müller et al., 2022; Hollmann et al., 2025). In that regime, the attention mechanism is the primary bottleneck, a point developed in the next section. Theorem 4.1 also justifies the practice of interpreting the final MLP layer as binned probabilities in PFNs; the MLP weights are likely closely connected to the sufficient statistics of each bin needed to represent the target density.

## 5. Generalizing Beyond Pretrain Sample Sizes

In this section, we provide an explanation based on our construction for what drives context size $n$-generalization—i.e., retaining reliable point prediction and uncertainty quantification beyond the pretraining sample size range—and what ultimately limits it beyond computational cost. We cast $n$-generalization as requiring an iterative solver that remains stable as the dimension of the underlying system varies with $n$, and we show that attention normalization promotes generalization by preconditioning. We then show that, even with preconditioning, generalizing to substantially larger $n$ becomes intrinsically more demanding and explain how deeper attention is essential to mitigate this difficulty.

In practice, PFN models are pretrained on varying $n$ by minimizing the loss over a *pretraining range* $n \in [n_{\min}, n_{\max}]$:

$$\vartheta = \arg\min_{\vartheta} -\mathrm{E}_n \mathrm{E}_{y, Z^{(0)}} \log q_\vartheta(y \mid Z^{(0)}),$$

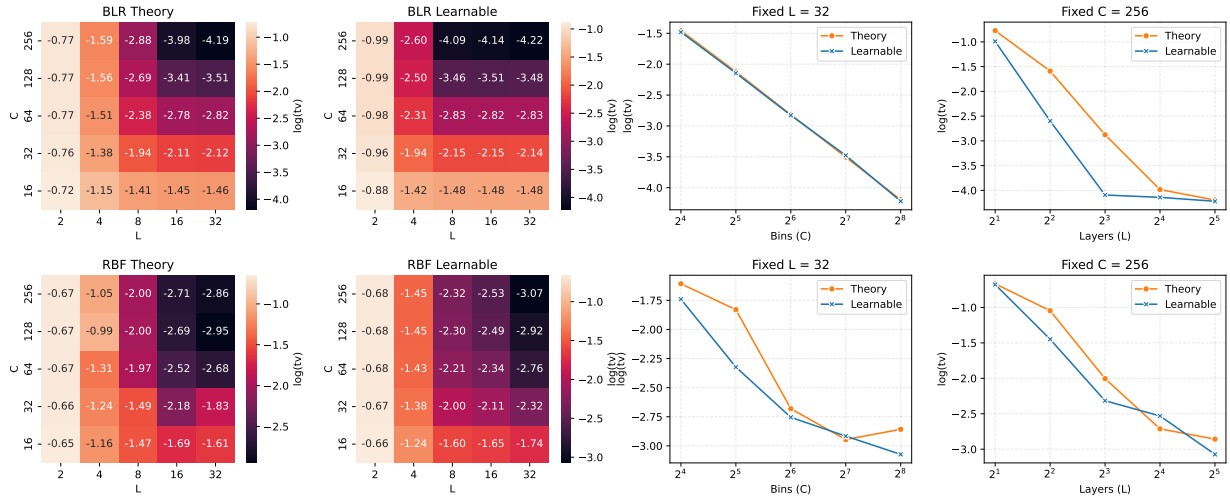

*Figure 2.* **Verifying Theorem 4.1.** $\log \mathrm{E}[\mathrm{TV}(p_{(a,b]}, q_\vartheta)]$ on the evaluation set as a function of depth $L \in \{2, 4, 8, 16, 32\}$ and bin count $C \in \{16, 32, 64, 128, 256\}$, for Bayesian linear regression (BLR) ($d = 5$, $n \in [128, 512]$) and GP regression with RBF kernel ($d = 2$, $n \in [64, 128]$). Left: heatmaps comparing *theory* and *learnable* parameterizations. Right: slices at fixed $L = 32$ (varying $C$) and fixed $C = 256$ (varying $L$), showing improvement with depth and bin resolution in both tasks.

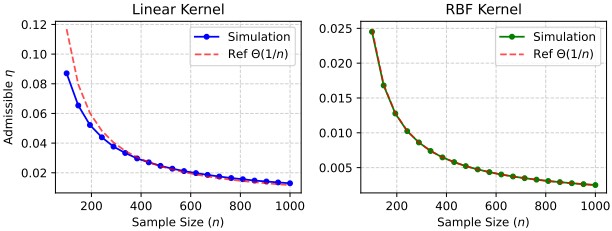

*Figure 3.* **Richardson step size varies with $n$ without normalization.** The upper bound of admissible step size of Richardson iteration solving (11), computed as (12), plotted against $n \in \{100 \times i : i \in [10]\}$ for $d = 16$ and $\sigma^2 = 0.2$. Results are averaged over 100 independent trials.

where the expectation over $n$ is taken with respect to a distribution on $\{n_{\min}, \dots, n_{\max}\}$ used in pretraining, such as discrete uniform. Models trained this way often exhibit nontrivial generalization beyond the pretraining range, although for sufficiently large $n$, they are sometimes outperformed by non-ICL supervised methods (Hollmann et al., 2023; Müller et al., 2025). This $n$-generalization problem is pressing because the pretraining range is constrained by computational cost, yet a single pretrained network is routinely evaluated at test time on context sizes $n$ outside this range. While several works propose strategies to scale PFNs to larger samples (Feuer et al., 2024; Wang et al., 2025; Qu et al., 2025), a mechanistic understanding of which architectural components drive $n$-generalization in PFNs—especially for both point prediction and PPD approximation—remains limited.

**Step size decreases with $n$.** Our construction renders these mechanisms explicit because the attention block $\mathrm{TF}_{\vartheta,L}$ is designed to implement an iterative solver for the PPD

moments. It must realize an iterative solver—with a single shared $\vartheta$—that approximately solves

$$(G + \sigma^2 I_n)u = Gv, \quad v \in \{Y, k_x\}, \tag{11}$$

for any $n$ within a fixed iteration $L$. If the learned parameters exactly match the data-generating parameters, the iteration converges as $L \to \infty$ for every $n$. However, $L$ is finite, so convergence is not guaranteed uniformly over $n$. Therefore, $\vartheta$ becomes tuned to the spectrum of $G$ typical for $n \in [n_{\min}, n_{\max}]$.

For simplicity, assume a shared step size $\eta^{(l)} = \eta$. The convergence of the recursion underlying $\mathrm{TF}_{\vartheta,L}$ requires

$$0 < \eta < 2/\{\lambda_1(G) + \sigma^2\} \tag{12}$$

(see Section A). Under the Gaussian design, for both linear and RBF kernels, the largest eigenvalue of $G$ grows linearly with $n$ as $\lambda_1(G) = \Theta(n)$ with high probability (see Lemma B.3-B.4). The step size theorem then follows, as illustrated in Figure 3.

**Theorem 5.1.** *With $x_i \overset{\mathrm{iid}}{\sim} \mathcal{N}(0, I_d/d)$ for fixed $d$, linear and RBF $\kappa$ yield a Richardson step size bound of order $1/n$.*

A step size $\hat{\eta}$ tuned to the range $n \in [n_{\min}, n_{\max}]$ must be small enough to satisfy (12) at $n_{\max}$, yet large enough to yield fast convergence within the finite depth $L$. Therefore, at inference, two failure modes could emerge: for $n' \ll n_{\min}$, $\hat{\eta}$ becomes overly conservative, and the recursion *under-converges* within $L$ steps. For $n' \gg n_{\max}$, $\hat{\eta}$ violates (12), and the recursion may *diverge*; see Figure 4 (left).

**Preconditioning ensures $n$-robustness.** A standard remedy is *Jacobi (diagonal) preconditioning*, which controls

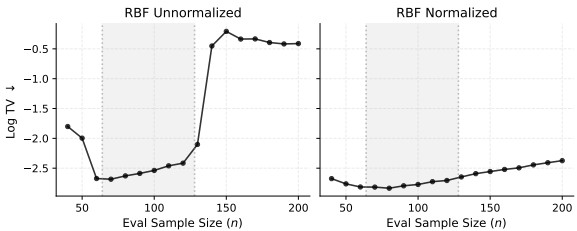

*Figure 4.* **Normalized attention enables sample size generalization (RBF, $d = 8$).** $\log \mathbb{E}[\text{TV}(p_{(a,b]}, q_\vartheta)]$ versus evaluation sample size $n' \in \{40, 50, \ldots, 200\}$ for *learnable* unnormalized $\text{TF}_{\vartheta,L}$ (left) and normalized $\text{TF}^{\text{pr}}_{\vartheta,L}$ (right), pretrained on $n \in [64, 128]$. Normalization yields stable performance across $n'$.

the spectrum of the iteration matrix uniformly over $n$ (Cutajar et al., 2016). Specifically, left-multiplying (11) by the inverse row-sum matrix $D^{-1}$ yields

$$D^{-1}(G + \sigma^2 I_n)u = D^{-1}Gv, \quad D = \text{diag}(G1_n), \quad (13)$$

with the same fixed points as (11). As $D^{-1}G$ is a row stochastic matrix, the maximum eigenvalue lies in $\lambda_1(D^{-1}(G + \sigma^2 I_n)) \in [1, 1 + \sigma^2]$ (see Lemma B.5). Consequently, a sufficient step size condition for convergence can be chosen independent of $n$. For the RBF kernel, a conservative range is $0 < \eta < 2/(1 + \sigma^2)$.

**Normalized attention implements preconditioning.** Jacobi preconditioning can be implemented with a minor modification of $\text{TF}_{\vartheta,L}$. The Richardson iteration for (13) can be written coordinate-wise as

$$
\begin{aligned}
u^{(l+1)}_{\text{pr}}(x) &= \left(1 - \frac{\eta}{s_x}\sigma^2\right) u^{(l)}_{\text{pr}}(x) \\
&+ \frac{\eta}{s_x} \sum_{i=1}^{n} \kappa(x_i, x)\left(v_i - u^{(l)}_{\text{pr}}(x_i)\right),
\end{aligned}
\quad (14)
$$

where $s_x = \sum_{i=1}^{n} \kappa(x_i, x)$ is the kernel aggregate for $x$. The preconditioner $s_x$ is readily obtained from *normalized* attention $\text{Attn}^{\text{na}}_{M,\kappa} : \mathbb{R}^{d' \times (n+1)} \to \mathbb{R}^{d' \times (n+1)}$, where the (masked) attention weights are normalized by their sum:

$$[\text{Attn}^{\text{na}}_{M,\kappa}(Z; V, K, Q)]_{:,j} = \sum_{i=1}^{n+1} \frac{\kappa(Kz_i, Qz_j)M_{ij}}{s_j} V z_i$$

with $s_j = \sum_{i'=1}^{n+1} \kappa(Kz_{i'}, Qz_j)M_{i'j}$. For strictly positive kernels where this normalization yields nonnegative weights (e.g., RBF), we define $\text{TF}^{\text{pr}}_{\vartheta,L}$ by replacing each head $\text{Attn}_{M,\kappa}$ in (10) with $\text{Attn}^{\text{na}}_{M,\kappa}$ and applying token-wise scaling $1/s_{x_j}$ to each token in the skip connection (the additive update implementing the $-\eta\sigma^2$ term), where $s_{x_j}$ is the attention normalizer for token $j$ (including the query token $x$). Under the same derivation as in the proof of Theorem 3.1, $\text{TF}^{\text{pr}}_{\vartheta,L}$ implements (14). This change is a lightweight local reweighting using attention output, rather

than an additional learned module that materially departs from standard transformer components.

**Larger $n$ demands more depth.** Regardless of Jacobi preconditioning, achieving a fixed accuracy with a finite-depth iterative solver becomes harder as $n$ grows. Under the optimal constant step size, the relative error of the Richardson iteration decays geometrically with the *convergence factor*

$$\rho^* = 1 - 2/\{\text{cond}(G + \sigma^2 I_n) + 1\} \in (0, 1),$$

where $\text{cond}(\cdot)$ denotes the condition number (see Section A). A larger $\rho^*$ close to 1 indicates slower error decay, requiring more iterations for a prescribed tolerance.

The $\rho^*$ captures the core difficulty of generalizing in $n$. For the RBF kernel on $\mathbb{R}^d$, the eigenvalues of the population integral operator decay geometrically (Rasmussen & Williams, 2005, Section 4.3.1.). Using standard connections between the integral operator and the empirical Gram (e.g., Burt et al. (2019)), one can show that $G$ becomes increasingly ill-conditioned with $n$ (see Lemma B.4). While a ridge term $\sigma^2 I_n$ lifts small eigenvalues, it does not stop growth with $n$ (Lemma B.5); Jacobi preconditioning similarly bounds the spectrum but leaves the order unchanged.

**Theorem 5.2.** *With $x_i \overset{\text{iid}}{\sim} \mathcal{N}(0, I_d/d)$ for fixed $d$ and RBF $\kappa$, $\text{cond}(D^{-1}(G + \sigma^2 I_n)) \overset{a.s.}{=} \Theta(n)$ for large $n$.*

Thus, the convergence factor $\rho^* \to 1$ as $n$ grows, posing two key implications. First, the attention block—a solver with a single weight—must be trained on a sufficiently wide range (large $n_{\max}$) so that the solver remains stable over a broader range of spectra. Second, because the convergence slows as $n$ increases, a deeper $L$ is essential to supply enough iteration budget to drive the solver error below a fixed tolerance. For the RBF kernel, its ill-conditioning makes the required depth grow essentially linearly in $n$.

**Theorem 5.3.** *Suppose the weight $\vartheta$ in $\text{TF}^{\text{pr}}_{\vartheta,L}$ encodes the optimal constant step size for the Richardson solving (13). For RBF $\kappa$, for a context $Z^{(0)}$ of size $n$, to achieve $\left\| \text{TF}^{\text{pr}}_{\vartheta,L}(Z^{(0)}) - (\mu(Z^{(0)}), \tau(Z^{(0)}))^\top \right\|_\infty < \epsilon$ for $\epsilon > 0$, it suffices that $L \gtrsim n \log(1/\epsilon)$ almost surely for large $n$.*

*Remark* 5.4 (Scalability of practical PFNs). Theorem 5.3 gives a sufficient condition for a deliberately constrained construction with fixed geometry and a single step size. By contrast, the *learnable* transformer used in Section 6 is more flexible, although it preserves key structural features of the construction: the $Q, K$ maps remain diagonal, and the sparsity pattern is inherited from the Richardson-style iteration. As shown in Section 6, this additional flexibility allows generalization to improve not only with depth $L$ but also with a wider pretraining sample size range, which exposes the model to a broader range of task eigenspectra. Thus, we view Theorem 5.3 as clarifying the role of depth in a simplified setting rather than claiming that its scaling law governs all practical PFNs.

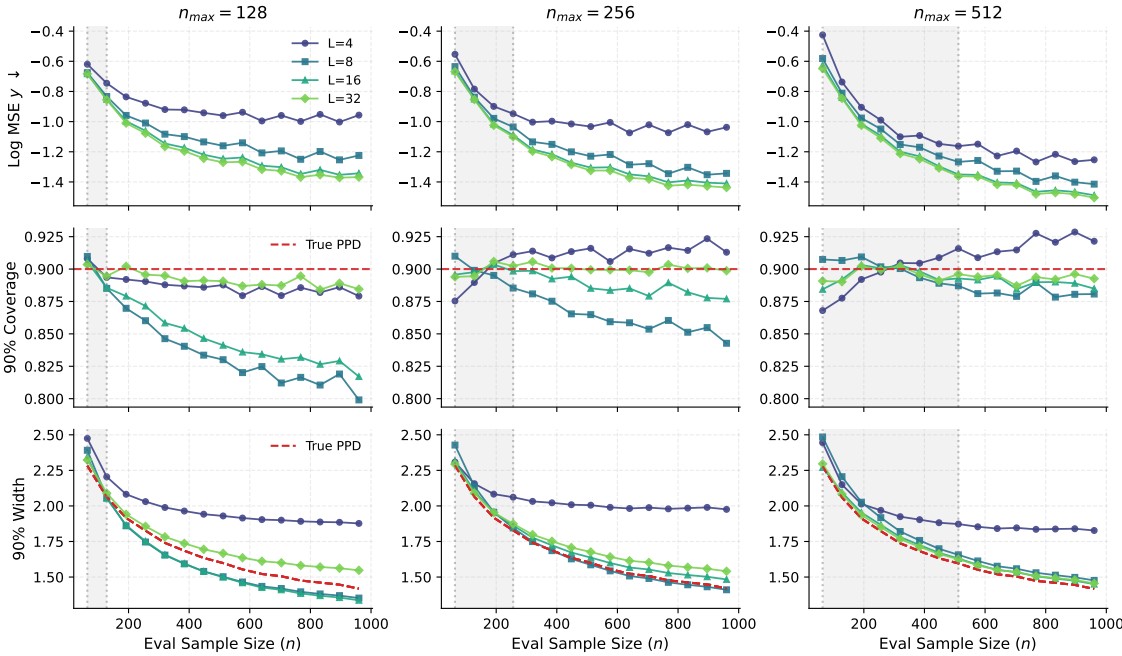

*Figure 5.* **Depth and pretraining range improve generalization quality (RBF, $d = 16$).** Prediction MSE, 90% interval coverage, and 90% interval width versus evaluation sample size for *learnable* normalized models with $C = 256$, depths $L \in \{4, 8, 16, 32\}$, and pretraining ranges $n \in [64, n_{\max}]$ with $n_{\max} \in \{128, 256, 512\}$. Red dashed curves denote the corresponding true PPD interval width/nominal coverage.

# 6. Numerical Studies

We validate our theoretical claims on two regression tasks: Bayesian linear regression (BLR; linear kernel) and nonlinear RBF regression (RBF kernel). For the unnormalized transformer $\mathrm{TF}_{\vartheta, L}$ in Theorem 3.1, we consider two parameterization modes: *theory*, where learnable weights are constrained as (23) and the nonzero entries of $K^{(l)} = Q^{(l)}$ are learned per layer, and *learnable*, where all nonzero parameters are learned independently. For the normalized transformer $\mathrm{TF}_{\vartheta, L}^{\mathrm{pr}}$, we consider only the *learnable* mode.

We focus on moderate dimensions, consistent with common GP practice: as $d$ grows, substantially more samples are typically needed for hyperparameter estimation, making training more costly and less stable. In GP-based surrogate modeling, applications are therefore often limited to 10-20 dimensions unless additional structure is imposed (Frazier, 2018; Moriconi et al., 2020).

**Pretraining.** For each pretraining instance, we draw a sample size $n \sim \mathrm{Unif}[n_{\min}, n_{\max}]$, and then sample $(\mathcal{D}_n, x, y)$ from the corresponding GP model. Given a choice of hyperparameters $(\sigma^2, \Sigma, \alpha, \ell)$, we draw inputs $x_1, \ldots, x_n \overset{\mathrm{iid}}{\sim} \mathcal{N}(0, I_d/d)$ and sample a latent function $\phi \sim \mathrm{GP}(0, \kappa)$, where $\kappa_{\mathrm{blr}}(x, x') = x^\top \Sigma x'$ and $\kappa_{\mathrm{rbf}}(x, x') = \alpha^2 \exp(-0.5 \|x - x'\|^2 / \ell^2)$. The responses are generated as $y_i \overset{\mathrm{ind}}{\sim} \mathcal{N}(\phi(x_i), \sigma^2)$. For the test token,

we independently draw $x \sim \mathcal{N}(0, I_d/d)$. Conditioned on $Z^{(0)} = \mathcal{D}_n \cup \{x\}$, we compute $\mu(Z^{(0)})$ and $\tau(Z^{(0)})$ and draw $y \sim \mathcal{N}(\mu(Z^{(0)}), \tau(Z^{(0)}))$ truncated to $(a, b]$, which is set based on Monte Carlo calibration over 2000 instances so that the probability outside the interval is 0.002. $Z^{(0)}$ is passed through the network to obtain a binned distribution $q_\vartheta(\cdot \mid Z^{(0)})$, and train by minimizing the (truncated) NLL.

**Inference.** At inference time, we draw an evaluation sample size $n' \sim \mathrm{Unif}[n'_{\min}, n'_{\max}]$ that may extend beyond the pretraining range; all other settings match pretraining unless stated otherwise (e.g., covariate shift experiment in Section E). Given $Z^{(0)}$, we let $p_{(a,b]}$ denote the true PPD truncated to $(a, b]$, and let $q_\vartheta$ be the piecewise-constant density on $(a, b]$ induced by the model's $C$-bin predictive probabilities.

Additional details on the choice of the MLP head and the hyperparameters $(\sigma^2, \Sigma, \alpha, \ell)$ can be found in Section D. Moreover, Section E contains additional simulation results with different choices of $(\sigma^2, \ell)$ and hierarchical GP regression with a finite discrete prior on random $(\sigma^2, \ell)$.

## 6.1. Validation of Theorem 4.1

We verify Theorem 4.1 by training *theory* and *learnable* $\mathrm{TF}_{\vartheta, L}$ over depths $L \in \{2, 4, 8, 16, 32\}$ and bin sizes $C \in \{16, 32, 64, 128, 256\}$. We use $d = 5$, $(n_{\min}, n_{\max}) = (128, 512)$ for BLR and $d = 2$, $(n_{\min}, n_{\max}) = (64, 128)$

for RBF, and evaluate on the same ranges. For each $(L, C)$, we report the average TV distance $\mathrm{E}[\mathrm{TV}(p_{(a,b)}, q_\vartheta)]$ on the evaluation set. In Figure 2, shown on a log–log scale, $\log \mathrm{TV}$ decreases linearly as the log of the bin size increases; it also decreases approximately according to a power law with respect to the log of the depth.

### 6.2. Normalized vs. Unnormalized Attention

We compare $n$-generalization of $\mathrm{TF}_{\vartheta,L}$ and $\mathrm{TF}_{\vartheta,L}^{\mathrm{pr}}$ on the RBF task. First, for $d = 8$ and pretraining range $(n_{\min}, n_{\max}) = (64, 128)$, we evaluate the *learnable* unnormalized $\mathrm{TF}_{\vartheta,L}$ and normalized $\mathrm{TF}_{\vartheta,L}^{\mathrm{pr}}$ with $L = 32$ and $C = 256$, reporting $\mathrm{E}[\mathrm{TV}(p_{(a,b)}, q_\vartheta)]$ for $n' \in \{40, 50, \ldots, 200\}$. Figure 4 shows that the unnormalized model's TV errors spike outside the pretraining range, while the normalized model remains stable.

### 6.3. Validation of Theorem 5.3

We examine how depth $L$ and pretraining range $n_{\max}$ affect $n$-generalization for $d = 16$, fixing $n_{\min} = 64$ and varying $n_{\max} \in \{128, 256, 512\}$ with $C = 256$ and $L \in \{4, 8, 16, 32\}$, evaluated at $n' \in \{64 \times i : i \in [16]\}$. We report metrics that align with downstream use: point prediction via $\mathrm{E}(y - m_{\vartheta,1})^2$ and uncertainty quantification via 90% coverage $\mathrm{P}\{y \in [l_\vartheta, u_\vartheta]\}$ and mean width $\mathrm{E}(u_\vartheta - l_\vartheta)$ (with the true PPD interval width as a baseline). To directly probe solver accuracy, we additionally report $\mathrm{E}[\mathrm{TV}(p_{(a,b)}, q_\vartheta)]$ and moment MSEs $\mathrm{E}(\mu - m_{\vartheta,1})^2$ and $\mathrm{E}(\tau + \mu^2 - m_{\vartheta,2})^2$, where $m_{\vartheta,1}(Z^{(0)}) := \hat{\mathrm{E}}(y \mid Z^{(0)}, \vartheta)$ and $m_{\vartheta,2}(Z^{(0)}) := \hat{\mathrm{E}}(y^2 \mid Z^{(0)}, \vartheta)$ are computed from the binned distribution $q_\vartheta$.

Figure 5 shows that generalization improves with larger $L$ and $n_{\max}$. Figure 10 in the Appendix shows that gains in prediction and interval metrics closely track corresponding reductions in solver convergence error (TV and moment errors) as $L$ and $n_{\max}$ increase, supporting the view that performance is primarily iteration-limited. However, consistent with Theorem 5.3, for fixed $L$ and $n_{\max}$, the error of the finite-iteration solver increases as the evaluation sample size grows. Additional simulations (e.g., $d \in \{4, 8\}$ and covariate shift) are deferred to Section E.

## 7. Discussion

This work provides a constructive account of how transformer-based PFNs can approximate PPDs in-context for GP regression, where the PPD is available in closed form. This setting allows us to make the target computation explicit, decompose the approximation error, and study how architectural choices such as depth, normalization, and output discretization affect distributional accuracy. While the results do not constitute a general theory of in-context learn-

ing for PFNs under arbitrary priors, our results provide a concrete foundation for a broad class of GP regression models, including Bayesian linear regression, nonparametric regression, and Gaussian state-space models.

Our contribution also provides insight into the expressive capabilities of PFNs in much broader settings. For example, in hierarchical GPs with a finite discrete prior over kernel and noise hyperparameters, the PPD becomes a finite mixture of componentwise GP predictive distributions; see Section E. This suggests a possible multi-head extension in which attention computes componentwise predictive moments in parallel while an additional readout module approximates the mixture weights. Another important class of models is latent GP models for non-Gaussian data, including GP classification and spatial/spatio-temporal models (Rue et al., 2009). Although exact PPDs are typically unavailable in closed form, approximate methods often rely on computation of latent GP posterior summaries that are closely related to GP regression predictive moments; see Chu & Ghahramani (2005, Section 4). Our decomposition suggests that attention mechanisms may be capable of implementing such iterative approximate posterior computations, while the output head maps the resulting summaries to a discretized approximation of the non-Gaussian PPD.

A limitation of our work is that the theoretical results are primarily constructive existence results, similar to Akyürek et al. (2023); Von Oswald et al. (2023), and do not imply that standard PFN pretraining necessarily discovers these mechanisms. Nevertheless, we believe that the construction is valuable from an interpretability perspective, as pretrained transformers are often difficult to interpret, and it is unclear whether the architecture can realize the PPD computation at all. We address this question by providing a simple and explicit configuration in which attention layers implement recursions for PPD moments, while a shallow MLP maps those summaries to binned predictive probabilities. The experiments partially bridge the gap between existence and learning by showing that learnable relaxations reproduce the predicted effects of depth, bin resolution, and normalization. These relaxations also motivate flexible but still interpretable model classes that help explain practical performance. Related work similarly interprets such relaxations as preconditioned gradient descent (Ahn et al., 2023) or anisotropic denoising (Rosu et al., 2026). That said, obtaining optimization guarantees for PFN pretraining and determining when trained transformers actually implement solver-like mechanisms remain important open problems.

### Software

The code is available at https://github.com/hun-learning94/transformer-uq.

## Acknowledgements

The authors thank the anonymous reviewers for their insightful comments and constructive feedback, which greatly improved the manuscript. The research of Changwoo Lee was partially supported by NIH R01ES035625, NSF IIS-2426762, and ONR N00014-24-1-2626.

## Impact Statement

This paper aims to advance the field of Machine Learning. While various societal impacts of our work are possible, none require specific emphasis here.

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

# A. Richardson Iteration

Consider a linear equation $Au^* = b$ where $A \in \mathbb{R}^{n \times n}$ is positive definite and $b \in \mathbb{R}^n$. Richardson iteration of solving the equation is given by

$$u^{(l+1)} = u^{(l)} + \eta(b - Au^{(l)}) = (I_n - \eta A)u^{(l)} + \eta b, \quad l \geq 0,$$

whose fixed point is $u^*$. We call $I_n - \eta A$ an iteration matrix and $\eta > 0$ a step size.

**Admissible Step Size for Convergence.** By subtracting $u^*$ in both sides, we see that $e^{(l)} := u^{(l)} - u^*$ evolves as

$$e^{(l+1)} = (I_n - \eta A)e^{(l)}.$$

Since $A \succ 0$, we can write the iteration matrix as $I_n - \eta A = Q(I_n - \eta \Lambda)Q^\top$ with the spectral decomposition of $A$. It follows that

$$\|e^{(l+1)}\|_2 \leq \max_i |1 - \eta\lambda_i(A)| \cdot \|e^{(l)}\|_2,$$

where $\lambda_1(A) \geq \cdots \geq \lambda_n(A) > 0$ are eigenvalues of $A$, and $\max_i |1 - \eta\lambda_i(A)|$ is the convergence factor of $I_n - \eta A$. Therefore, $u^{(l)} \to u^*$ if and only if $\max_i |1 - \eta\lambda_i(A)| < 1$, which translates to a condition on $\eta$ as

$$0 < \eta < \frac{2}{\lambda_1(A)}.$$

Note that the smaller the convergence factor, the faster the convergence.

**Optimal Step Size.** The optimal step size $\eta^*$ minimizes the convergence factor $\max_i |1 - \eta\lambda_i(A)|$. Since $\max_i |1 - \eta\lambda_i(A)| = \max\{|1 - \eta\lambda_1(A)|, |1 - \eta\lambda_n(A)|\}$, $\eta^*$ satisfies $-1 + \eta^*\lambda_n(A) = 1 - \eta^*\lambda_1(A)$; hence, the optimal step size and the corresponding optimal convergence factor become

$$\eta^* = \frac{2}{\lambda_1(A) + \lambda_n(A)}, \quad \rho^* = 1 - \frac{2}{(\lambda_1/\lambda_n)(A) + 1}$$

where $(\lambda_1/\lambda_n)(A)$ is the condition number of $A$.

**Convergence Rate.** Suppose $0 < \eta < 1/\lambda_1(A)$. Then the convergence factor becomes $\rho := \max_i |1 - \eta\lambda_i(A)| = 1 - \eta\lambda_n(A)$, and we can write, taking $u^{(0)} = 0$,

$$\|u^{(L)} - u^*\|_2 / \|u^*\|_2 \leq (1 - \eta\lambda_n(A))^L = (1 - (1 - \rho))^L \leq \exp\left(-(1 - \rho)L\right).$$

**Preconditioning.** Let $D \in \mathbb{R}^{n \times n}$ be positive definite. A preconditioned Richardson iteration solves $D^{-1}Au^* = D^{-1}b$, and is given by

$$u^{(l+1)} = u^{(l)} + \eta D^{-1}(b - Au^{(l)}) = (I_n - \eta D^{-1}A)u^{(l)} + \eta D^{-1}b, \quad l \geq 0.$$

It is clear that the above has the same fixed point as the unconditioned one. It naturally follows that

$$0 < \eta < \frac{2}{\lambda_1(D^{-1}A)}, \quad \eta^* = \frac{2}{\lambda_1(D^{-1}A) + \lambda_n(D^{-1}A)}, \quad \rho^* = 1 - \frac{2}{(\lambda_1/\lambda_n)(D^{-1}A) + 1}.$$

Note that $D^{-1}A$ and $D^{-1/2}AD^{-1/2}$ are similar matrices, so they have the same characteristic polynomials, hence the same eigenvalues. Hence, the above quantities can be expressed in terms of $\lambda_j(D^{-1/2}AD^{-1/2})$, $j \in \{1, n\}$ as well.

# B. Auxiliary Lemmas and Theorems

## B.1. Approximation based on Discretization

**Lemma B.1.** *Let $p(x)$ be a $L_{\mathrm{Lip}}$-Lipschitz continuous density supported on $\mathbb{R}$. Fix a partition $\Gamma = \{a = \gamma_1 < \gamma_2 < \cdots < \gamma_{C+1} = b\}$ of an interval $(a, b]$ such that $\mathbb{P}\{x \notin (a, b]\} = \varepsilon$ for $\varepsilon > 0$. Let $\Delta_c := \gamma_{c+1} - \gamma_c$ and $|\Gamma| := \max_{1 \leq c \leq C} \Delta_c$. Define $\tilde{p}(x) = \sum_{c=1}^{C} \Delta_c^{-1} 1_{[\gamma_c, \gamma_{c+1})}(x) \int_{\gamma_c}^{\gamma_{c+1}} p(x)dx/(1 - \varepsilon)$. Then $\mathrm{TV}(p, \tilde{p}) \leq L_{\mathrm{Lip}}|\Gamma|(b - a)/2 + \varepsilon$.*

*Proof.* Note that $2\,\mathrm{TV}(p,\tilde{p}) = \int |p - \tilde{p}|dx$ where

$$
\int |p - \tilde{p}|dx \leq \int_{(a,b]} |p - \tilde{p}|dx + \int_{(a,b]^c} |p - \tilde{p}|dx
$$

$$
\leq \int_{(a,b]} |p - (1-\varepsilon)\tilde{p}|dx + \int_{(a,b]} |(1-\varepsilon)\tilde{p} - \tilde{p}|dx + \int_{(a,b]^c} |p - \tilde{p}|dx
$$

$$
= \sum_{c=1}^{C} \int_{\gamma_c}^{\gamma_{c+1}} |p(x) - p(x_c)|dx + 2\varepsilon
$$

where $x_c \in [\gamma_c, \gamma_{c+1}]$ is the point in the interval whose density is equal to the mean value of density in the interval. By $L_{\mathrm{Lip}}$-Lipschitz,

$$
\sum_{c=1}^{C} \int_{\gamma_c}^{\gamma_{c+1}} |p(x) - p(x_c)|dx \leq \sup_{|x-x'|\leq|\Gamma|} |p(x) - p(x')| \sum_{c=1}^{C} \Delta_c \leq L_{\mathrm{Lip}}|\Gamma|(b-a),
$$

which completes the proof. $\qquad\square$

**Lemma B.2.** *Fix $\theta \in \mathbb{R}^k$ and let $p(\cdot \mid \theta)$ be an exponential-family density on $\mathbb{R}$:*

$$
p(y \mid \theta) = \exp\left(\langle \theta, T(y)\rangle - A(\theta) + \log h(y)\right).
$$

*Choose $a < b$ such that $p(\cdot \mid \theta)$ is continuous and positive on $[a,b]$ and $\int_a^b p(y \mid \theta)dy = 1 - \epsilon$ for $\epsilon > 0$. Define $p_{(a,b]}(y \mid \theta) := p(y \mid \theta)\mathbf{1}_{(a,b]}(y)/(1-\epsilon)$. For each $C \geq 1$, let $\Gamma^{(C)} = \{a = \gamma_1 < \gamma_2 < \cdots < \gamma_{C+1} = b\}$ be a partition with mesh size $|\Gamma^{(C)}| := \max_{c \in [C]}(\gamma_{c+1} - \gamma_c)$ satisfying $|\Gamma^{(C)}| \to 0$ as $C \to \infty$. Let $\xi_c := (\gamma_c + \gamma_{c+1})/2$ be the midpoint for $c \in [C]$ and let $\Delta_c := \gamma_{c+1} - \gamma_c$. Define two pdfs on $(a,b]$:*

$$
p_C(y \mid \theta) := \sum_{c=1}^{C} \mathbf{1}_{y \in (\gamma_c, \gamma_{c+1}]} \Delta_c^{-1} \int_{\gamma_c}^{\gamma_{c+1}} p_{(a,b]}(t \mid \theta)dt
$$

$$
\hat{p}_C(y \mid \theta) := \sum_{c=1}^{C} \mathbf{1}_{y \in (\gamma_c, \gamma_{c+1}]} \Delta_c^{-1} \mathrm{sm}(\ell)_c, \quad \ell_c := \langle \theta, T(\xi_c)\rangle + \log h(\xi_c) + \log \Delta_c.
$$

*Letting $\omega_p(\delta) := \sup\{|p(y \mid \theta) - p(y' \mid \theta)| : y, y' \in (a,b], |y - y'| \leq \delta\}$, for every $C$,*

$$
\mathrm{TV}\left(p_C, \hat{p}_C\right) \leq \frac{(b-a)}{1-\epsilon}\omega_p\left(\frac{|\Gamma^{(C)}|}{2}\right).
$$

*In particular, if $p(\cdot \mid \theta)$ is $L_{\mathrm{Lip}}$-Lipschitz on $(a,b]$, then*

$$
\mathrm{TV}\left(p_C, \hat{p}_C\right) \leq \frac{(b-a)}{4(1-\epsilon)}L_{\mathrm{Lip}}|\Gamma^{(C)}|.
$$

*Proof.* Define two pmfs on $\{1, \ldots, C\}$:

$$
q_C(c \mid \theta) := \int_{\gamma_c}^{\gamma_{c+1}} p_{(a,b]}(y \mid \theta)dy,
$$

$$
\hat{q}_C(c \mid \theta) := \mathrm{sm}(\ell)_c,
$$

and note that $\mathrm{TV}\left(p_C, \hat{p}_C\right) = \mathrm{TV}\left(q_C, \hat{q}_C\right)$ since $\|p_C - \hat{p}_C\|_{L^1(a,b]} = \sum_{c=1}^{C} |q_C(c \mid \theta) - \hat{q}_C(c \mid \theta)|$. Since $A(\theta)$ is an additive constant of the log-density independent of $c$, it cancels under $\mathrm{sm}$. Therefore,

$$
\hat{q}_C(c \mid \theta) = \frac{p(\xi_c \mid \theta)\Delta_c}{\sum_{j=1}^{C} p(\xi_j \mid \theta)\Delta_j}.
$$

Define the following: $g_c := \int_{\gamma_c}^{\gamma_{c+1}} p(y \mid \theta) dy$, $\hat{g}_c := p(\xi_c \mid \theta) \Delta_c$, $Z := \sum_{c=1}^C g_c = 1 - \epsilon$, and $\hat{Z} := \sum_{c=1}^C \hat{g}_c$. Note that $\mathrm{TV}(q_C, \hat{q}_C) = \|q_C - \hat{q}_C\|_1 / 2$ where

$$\|q_C - \hat{q}_C\|_1 \leq \sum_{c=1}^C \left| \frac{g_c}{Z} - \frac{\hat{g}_c}{\hat{Z}} \right| = \sum_{c=1}^C \left| \frac{g_c}{Z} - \frac{\hat{g}_c}{Z} + \frac{\hat{g}_c}{Z} - \frac{\hat{g}_c}{\hat{Z}} \right|$$

$$\leq \frac{1}{Z} \sum_{c=1}^C |g_c - \hat{g}_c| + \frac{|\hat{Z} - Z|}{Z\hat{Z}} \sum_{c=1}^C \hat{g}_c$$

$$\leq \frac{2}{Z} \sum_{c=1}^C |g_c - \hat{g}_c|.$$

The last inequality holds since $|\hat{Z} - Z| = |\sum_{c=1}^C (\hat{g}_c - g_c)| \leq \sum_{c=1}^C |\hat{g}_c - g_c|$. Now, consider the term $|g_c - \hat{g}_c|$. Since $p$ is $L_{\mathrm{Lip}}$-Lipschitz:

$$|g_c - \hat{g}_c| = \left| \int_{\gamma_c}^{\gamma_{c+1}} (p(y \mid \theta) - p(\xi_c \mid \theta)) dy \right| \leq \int_{\gamma_c}^{\gamma_{c+1}} |p(y \mid \theta) - p(\xi_c \mid \theta)| dy \leq L_{\mathrm{Lip}} \int_{\gamma_c}^{\gamma_{c+1}} |y - \xi_c| dy.$$

Since $\xi_c$ is the midpoint of $[\gamma_c, \gamma_{c+1}]$, the integral $\int_{\gamma_c}^{\gamma_{c+1}} |y - \xi_c| dy$ represents the sum of the areas of two identical right-angled triangles, each with base $\Delta_c/2$ and height $\Delta_c/2$. Thus:

$$\int_{\gamma_c}^{\gamma_{c+1}} |y - \xi_c| dy = 2 \cdot \left( \frac{1}{2} \cdot \frac{\Delta_c}{2} \cdot \frac{\Delta_c}{2} \right) = \frac{\Delta_c^2}{4}.$$

Substituting this back into the sum:

$$\sum_{c=1}^C |g_c - \hat{g}_c| \leq L_{\mathrm{Lip}} \sum_{c=1}^C \frac{\Delta_c^2}{4} \leq \frac{L_{\mathrm{Lip}}}{4} \left( \max_c \Delta_c \right) \sum_{c=1}^C \Delta_c = \frac{L_{\mathrm{Lip}}}{4} |\Gamma^{(C)}| (b - a).$$

For the general continuous case (where Lipschitz continuity is not assumed), we retain the bound using the modulus of continuity $\omega_p(|\Gamma^{(C)}|/2)$, noting that $|g_c - \hat{g}_c| \leq \Delta_c \omega_p(|\Gamma^{(C)}|/2)$. Combining these estimates with the TV bound yields the stated result. $\qquad \square$

## B.2. Spectral Scaling of Gram Matrix

Throughout this section, we consider $x_i \overset{\mathrm{iid}}{\sim} \mathcal{N}(0, I_d/d)$ for $i \in [n]$ where $n > d$ for a fixed $d$.

**Lemma B.3** (Linear kernel). *Let $G := XX^\top$, a rank $d$ matrix, and $\lambda_1 \geq \cdots \geq \lambda_d$ be its $d$ nonzero eigenvalues. Then $\lambda_j = \left( 1 + O_{\mathbb{P}}(n^{-1/2}) \right) n/d$ for $j \in [d]$.*

*Proof.* Note that $X = Z/\sqrt{d}$ where $Z_{ij} \overset{\mathrm{iid}}{\sim} \mathcal{N}(0, 1)$. Letting $s_1 \geq \cdots \geq s_d$ be the singular values of $Z$, it holds that $s_j = \sqrt{d\lambda_j}$. By Boucheron et al. (2013, Theorem 5.6), if $z \sim \mathcal{N}(0, I_{d'})$, $f : \mathbb{R}^{d'} \to \mathbb{R}$ is $L$-Lipschitz, then for any $t \geq 0$,

$$\mathbb{P}\{f(z) \geq \mathbb{E}f(z) + t\} \leq e^{-t^2/(2L^2)}.$$

The mapping $Z \mapsto s_1(Z)$ is 1-Lipschitz as a function of $\mathrm{vec}(Z)$ with respect to $\ell_2$ norm, since $s_1(Z) = \max_{\|v\|_2=1} \|Zv\|_2 = \|Z\|_2$ and

$$|s_1(Z) - s_1(Z')| = |\|Z\|_2 - \|Z'\|_2| \leq \|Z - Z'\|_2 \leq \|Z - Z'\|_F = \|\mathrm{vec}(Z) - \mathrm{vec}(Z')\|_2$$

where $\|Z\|_F$ denotes the Frobenius norm. Moreover, since $\max_j |s_j(Z) - s_j(Z')| \leq \|Z - Z'\|_2$, $s_1, \cdots, s_d$ are all 1-Lipschitz (see Horn & Johnson (2012, Theorem 7.4.9.1)). On the other hand, by Davidson & Szarek (2001, Theorem 2.13),

$$\sqrt{n} - \sqrt{d} \leq \mathbb{E}s_d(Z) \leq \mathbb{E}s_1(Z) \leq \sqrt{n} + \sqrt{d}.$$

Combining these two facts, we have, for $j \in [d]$,

$$\lambda_j \in \left[ \frac{n}{d} \left( 1 - \sqrt{\frac{d}{n}} - \frac{t}{\sqrt{n}} \right)^2, \frac{n}{d} \left( 1 + \sqrt{\frac{d}{n}} + \frac{t}{\sqrt{n}} \right)^2 \right]$$

on the event $\mathcal{E}$ with $\mathbb{P}(\mathcal{E}) \geq 1 - 2e^{-t^2/2}$. Take $\epsilon_n := \frac{\lambda_j}{n/d} - 1$. Then $\sqrt{n}\epsilon_n = O_{\mathbb{P}}(1)$; hence, $\lambda_j = \left( 1 + O_{\mathbb{P}}(n^{-1/2}) \right) n/d$. $\square$

**Lemma B.4** (RBF kernel). *Define $G \in \mathbb{R}^{n \times n}$ as $G_{ij} = \exp\left( -0.5\|x_i - x_j\|^2 \right)$. Then $\lambda_1(G) \gtrsim n$ and there exists $\gamma > 0$ where*

$$\frac{\lambda_1(G)}{\lambda_n(G)} \gtrsim \frac{\exp(\gamma n^{1/d})}{n^{2+(d-1)/d}}.$$

*almost surely for any large enough $n$.*

*Proof.* The analytic eigenstructure of the Gaussian (RBF) kernel under a Gaussian input measure is available in closed form. In one dimension, Zhu et al. (1997) showed that the Mercer eigenvalues decay geometrically: $\mu_i \asymp \beta^i$ for $i = 0, 1, 2, \ldots$ with some $\beta \in (0, 1)$. This form extends to isotropic multivariate Gaussians because, for a product Gaussian measure and an isotropic RBF kernel, the eigenfunctions factorize across coordinates, and the eigenvalues take a product form with combinatorial multiplicities; see Rasmussen & Williams (2005, Section 4.3.1). In particular, when the spectrum is grouped by total degree, the distinct eigenvalues still decay geometrically in that degree, while their multiplicities grow polynomially. This geometric decay of the population spectrum can be transferred to the empirical spectrum of the Gram matrix $G$ via a Mercer truncation argument: approximating the kernel by its first $T < n$ eigencomponents yields a rank $T$ Gram matrix $G_{<T}$, and the $(T+1)$-st (hence smallest) sample eigenvalues is upper bounded through trace bounds and Markov inequality, following the technique in Burt et al. (2019).

Let $\kappa(x, x') = \exp\left( -0.5\|x - x'\|^2 \right)$ be the RBF kernel. By Mercer's theorem, the spectral decomposition is $\kappa(x, x') = \sum_{j \geq 1} \mu_j \phi_j(x) \phi_j(x')$ where $\{\phi_j\}$ are orthonormal basis and $\{\mu_j\}$ are eigenvalues.

**The largest eigenvalue.** Since $G \succ 0$, $\lambda_1(G) = \max_{x \neq 0} x^\top G x / x^\top x$, hence

$$\lambda_1(G) > \frac{1^\top G 1}{1^\top 1} = 1 + \frac{n-1}{n(n-1)} \sum_{i \neq j} \kappa(x_i, x_j).$$

Note that by the strong law of U-statistics, the average of off diagonal elements converges:

$$\frac{\sum_{i \neq j} \kappa(x_i, x_j)}{n(n-1)} \overset{a.s.}{\to} \rho := \mathbb{E}_{x, x'} \kappa(x, x') \in (0, 1).$$

Therefore, $\lambda_1(G) \gtrsim n$ for any sufficiently large $n$ almost surely.

**The smallest eigenvalue.** Fix $1 \leq T < n$ and define a truncated kernel $\kappa_{\leq T}(x, x') := \sum_{t=1}^{T} \mu_t \phi_t(x) \phi_t(x')$ with the Gram matrix $G_{\leq T} \in \mathbb{R}^{n \times n}$ such that $(G_{\leq T})_{ij} = \kappa_{\leq T}(x_i, x_j)$. We write $G = G_{\leq T} + R_{>T}$ where $R_{>T}$ is the residual matrix. Then we have

$$\lambda_n(G) \leq \lambda_{T+1}(G) \leq \lambda_{T+1}(G_{\leq T}) + \lambda_1(R_{>T}) = \lambda_1(R_{>T}) \leq \operatorname{tr}(R_{>T})$$

where the first inequality is since $T < n$, the second is from Weyl's inequality (if $A, B \in \mathbb{R}^{n \times n}$ are symmetric, then $\lambda_i(A + B) \in \left[ \lambda_i(A) + \lambda_n(B), \lambda_i(A) + \lambda_1(B) \right]$), the equality is because $G_{\leq T}$ has a rank $T$, and the final inequality holds since $R_{>T} \succ 0$. Now,

$$\operatorname{tr}(R_{>T}) = \sum_{i=1}^{n} \left( \kappa(x_i, x_i) - \kappa_{\leq T}(x_i, x_i) \right) = \sum_{i=1}^{n} \left( \sum_{t > T} \mu_t \phi_t(x_i)^2 \right),$$

and since $E\phi_t(x)^2 = \|\phi_t\|_{L^2}^2 = 1$ from $\phi_t$ being an orthonormal basis, we have $E\,\mathrm{tr}(R_{>T}) = n\sum_{t>T}\mu_t$. Then we can, as in the proof of Burt et al. (2019, Theorem 4), build a probabilistic bound using Markov's inequality: for a sequence $a_n$,

$$P\left(\lambda_n(G) > a_n n \sum_{t>T}\mu_t\right) \le P\left(\mathrm{tr}(R_{>T}) > a_n n \sum_{t>T}\mu_t\right) \le \frac{E\,\mathrm{tr}(R_{>T})}{a_n n \sum_{t>T}\mu_t} = \frac{1}{a_n}.$$

If we take $a_n = n^{1+\epsilon}$ for any $\epsilon > 0$ such that $\sum_{n\ge 1} 1/a_n$ is summable, then by the Borel-Cantelli lemma, we have $\lambda_n(G) \le n^{2+\epsilon}\sum_{t>T}\mu_t$ almost surely for any sufficiently large $n$.

We now use the following fact (Zhu et al., 1997) to get an explicit form of the tail sum: if $x \sim \mathcal{N}(0,\sigma^2)$ and $\kappa^{1D}(x,x') = \exp\left(-0.5(x-x')^2/\ell^2\right)$, then the eigenvalues $\{\mu_i\}$ of $\kappa^{1D}$ decay geometrically in the following form:

$$\mu_i^{1D} = \sqrt{\frac{2a}{a+b+\sqrt{a^2+2ab}}}\left(\frac{b}{a+b+\sqrt{a^2+2ab}}\right)^i$$

where $1/a = 4\sigma^2$ and $1/b = 2\ell^2$. This result transfers directly to the multivariate isotropic Gaussian measure, since the eigenvalues and eigenfunctions are also a product of the same univariate Gaussian. In isotropic case, an eigenfunction in $d$-dimension is a tensor product of $d$ number of one-dimensional eigenfunctions, so it is indexed by a multi index $(i_1, i_2, \cdots, i_d)$. Accordingly, eigenfunctions with the same value of $i_1 + \cdots + i_d$ (degree) share the same eigenvalue. Hence, the multiplicity of $\mu_k$ equals the number of possible ways to distribute $k$ balls into $d$ bins.

Specifically, let $\tilde{\mu}^{(k)}$ be an eigenvalue of $\kappa(x,x')$ whose degree is $k$ and has a multiplicity $m_k = \binom{k+d-1}{d-1}$. Each $\tilde{\mu}^{(k)}$ is a product of $\mu_i^{1D}$ in a multi index $(i_1, i_2, \cdots, i_d)$ such that $i_1 + \cdots + i_d = k$:

$$\tilde{\mu}^{(k)} = \left(\frac{2a}{a+b+\sqrt{a^2+2ab}}\right)^{d/2}\left(\frac{b}{a+b+\sqrt{a^2+2ab}}\right)^k =: \alpha^{d/2}\beta^k.$$

Let $T_{<M} := |\{(i_1, \cdots, i_d) : i_1 + \cdots + i_d < M\}|$ be the total number of eigenvalues whose degree $k$ is less than $M$. Then

$$T_{<M} = \sum_{k=0}^{M-1}\binom{k+d-1}{d-1} = \binom{M+d-1}{d}.$$

The tail sum of eigenvalues of degree larger or equal to $M$ is

$$\sum_{t>T_{<M}}\mu_t = \sum_{k\ge M}\binom{k+d-1}{d-1}\tilde{\mu}^{(k)} = \alpha^{d/2}\sum_{k\ge M}\binom{k+d-1}{d-1}\beta^k.$$

Fix $q \in (\beta, 1)$. Writing $\gamma_k := \binom{k+d-1}{d-1}\beta^k$, we choose $M$ large enough so that

$$\frac{\gamma_{k+1}}{\gamma_k} = \beta\left(1 + \frac{d-1}{k+1}\right) \le \beta\left(1 + \frac{d-1}{M+1}\right) < q < 1, \quad \forall k. \tag{15}$$

Then $\sum_{k\ge M}\gamma_k \le \gamma_M \sum_{t\ge 0} q^t = \gamma_M/(1-q)$; hence

$$\sum_{t>T_{<M}}\mu_t \le \alpha^{d/2}\binom{M+d-1}{d-1}\beta^M\frac{1}{1-q} = C\beta^M M^{d-1}$$

where $C$ is a constant that only depends on $\alpha$, $\beta$, $q$, and $d$, since $\binom{M+d-1}{d-1} \sim M^{d-1}/(d-1)!$ (note that $a_M \sim b_M$ means $a_M/b_M \to 1$ as $M \to \infty$).

For $n$, choose a small $a > 0$ such that after letting $M = \lfloor an^{1/d}\rfloor$, we have $n > T_{<M}$. Since $T_{<M} \sim M^d/d!$, one such choice could be $a \lesssim (d!)^{1/d}$. We assume $n$ is large enough so that $M$ satisfies (15). Then

$$\lambda_n(G) \le n^{2+\epsilon}\sum_{t>T_{<M}}\mu_t \le Cn^{2+\epsilon}\beta^M M^{d-1} \le Cn^{2+\epsilon}\beta^{an^{1/d}}(an^{1/d})^{d-1} = C'n^{2+\epsilon+\frac{d-1}{d}}\exp\left(-\gamma n^{1/d}\right)$$

where $\gamma = a\log(1/\beta) > 0$ (note that $0 < \beta < 1$) and $C'$ is a constant.

**Condition number.** Combining $\lambda_1(G) \gtrsim n$ and $\lambda_n(G) \lesssim n^{2+\epsilon+\frac{d-1}{d}}\exp\left(-\gamma n^{1/d}\right)$, we have

$$\frac{\lambda_1(G)}{\lambda_n(G)} \gtrsim \frac{\exp(\gamma n^{1/d})}{n^{1+\epsilon+(d-1)/d}}.$$

almost surely for any large enough $n$, which completes the proof by taking $\epsilon = 1$. $\qquad\square$

**Lemma B.5** (RBF kernel with ridge). *Define $G \in \mathbb{R}^{n\times n}$ as $G_{ij} = \exp\left(-0.5\|x_i - x_j\|^2\right)$. Let $\sigma^2 > 0$. Then*

$$\frac{\lambda_1(G) + \sigma^2}{\lambda_n(G) + \sigma^2} = \Theta(n)$$

*almost surely for any large enough $n$.*

*Proof.* Note that $\lambda_1(G) \le \mathrm{tr}(G) = n$ and $\lambda_n(G) \le \mathrm{tr}(G)/n = 1$. For large enough $n$, almost surely we have

$$\frac{1 + (n-1)\rho + \sigma^2}{1 + \sigma^2} \le \frac{\lambda_1(G) + \sigma^2}{\lambda_n(G) + \sigma^2} \le \frac{n + \sigma^2}{\sigma^2}$$

where $\rho \in (0,1)$ is defined in the proof of Lemma B.4. $\qquad\square$

**Lemma B.6** (RBF kernel with ridge and row normalization). *Define $G \in \mathbb{R}^{n\times n}$ as $G_{ij} = \exp\left(-0.5\|x_i - x_j\|^2\right)$. Let $D := \mathrm{diag}(s_1, \cdots, s_n)$ where $s_i := \sum_{j=1}^n G_{ij}$. Then*

$$\frac{\lambda_1(D^{-1}(G + \sigma^2 I_n))}{\lambda_n(D^{-1}(G + \sigma^2 I_n))} = \Theta(n)$$

*almost surely for any large enough $n$.*

*Proof.* We write $A \sim B$ to denote that $A$ and $B$ are similar matrices, sharing the same eigenvalues. Then $D^{-1}(G + \sigma^2 I_n) \sim D^{-1/2}(G + \sigma^2 I_n)D^{-1/2} =: \tilde{A} + \sigma^2 D^{-1}$. Since $D^{-1}G$ is a row stochastic matrix, we have $\lambda_1(\tilde{A}) = \lambda_1(D^{-1}G) = 1$. Also, by Weyl's inequality, $\lambda_1(\tilde{A} + \sigma^2 D^{-1}) \le 1 + \sigma^2/s_{\min} \le 1 + \sigma^2$ since $s_i \ge 1$ for any $i \in [n]$. Therefore,

$$\lambda_1(\tilde{A} + \sigma^2 D^{-1}) \in [1, 1 + \sigma^2].$$

On the other hand, $\lambda_n(\tilde{A} + \sigma^2 D^{-1}) \ge \sigma^2/s_{\max}$, and for a basis vector $e_i$, $\lambda_n(\tilde{A} + \sigma^2 D^{-1}) \le \frac{e_i^\top (\tilde{A}+\sigma^2 D^{-1})e_i}{e_i^\top e_i} \le \frac{1+\sigma^2}{s_i}$. Since this holds for any $i \in [n]$, we have

$$\lambda_n(\tilde{A} + \sigma^2 D^{-1}) \in [\sigma^2/s_{\max}, (1+\sigma^2)/s_{\max}].$$

Combined,

$$\frac{\lambda_1(D^{-1}(G + \sigma^2 I_n))}{\lambda_n(D^{-1}(G + \sigma^2 I_n))} \in \left[\frac{s_{\max}}{\sigma^2 + 1}, \frac{s_{\max}(\sigma^2 + 1)}{\sigma^2}\right].$$

Trivially, $s_{\max} \le n$ since all elements are in $[0,1]$. For any $i \in [n]$, consider

$$\frac{s_i}{n} = \frac{1}{n} + \frac{n-1}{n}\frac{\sum_{j\ne i}\kappa(x_i, x_j)}{n-1}.$$

Conditioned on $x_i$, $\kappa(x_i, x_j)$ are iid with mean $\mathbb{E}_x\kappa(x_i, x) \in (0,1)$. Therefore, conditionally on $x_i$, by the law of large number,

$$\mathrm{P}\left(\frac{\sum_{j\ne i}\kappa(x_i, x_j)}{n-1} \to \mathbb{E}_x\kappa(x_i, x) \mid x_i\right) = 1.$$

By taking expectation with respect to $x_i$, the convergence holds unconditionally. Note that $\mathbb{E}_x\kappa(x_i, x)$ and $\mathbb{E}_{x,x'}\kappa(x, x')$ exist as a Gaussian integral. Therefore, for large enough $n$, writing $\rho := \mathbb{E}_{x,x'}\kappa(x, x') \in (0,1)$,

$$s_{\max} \ge s_i = 1 + (n-1)\rho$$

almost surely. Combined, $s_{\max} = \Theta(n)$ almost surely, which completes the proof. $\qquad\square$

## B.3. Population Optimizer of NLL

**Theorem B.7.** *For each $Z^{(0)}$, define the truncated conditional density*

$$p_{(a,b]}(y \mid Z^{(0)}) := p(y \mid Z^{(0)})\mathbf{1}_{\{y \in (a,b]\}} \Big/ \int_a^b p(t \mid Z^{(0)})dt,$$

*assuming $\int_a^b p(t \mid Z^{(0)})dt > 0$ almost surely. Consider the class of piecewise-constant conditional densities supported on $(a, b]$:*

$$\mathcal{Q} := \left\{ q(\cdot \mid Z^{(0)}) : q(y \mid Z^{(0)}) = \sum_{c=1}^C \mathbf{1}_{\{y \in [\gamma_c, \gamma_{c+1})\}} \Delta_c^{-1} q_c(Z^{(0)}), \ q_c(Z^{(0)}) \geq 0, \ \sum_{c=1}^C q_c(Z^{(0)}) = 1 \right\}.$$

*Define the truncated population negative log-likelihood*

$$\mathcal{L}_{\mathrm{tr}}(q) := \mathbb{E}_{Z^{(0)}} \mathbb{E}_{y \sim p_{(a,b]}(\cdot \mid Z^{(0)})} \left[ -\log q(y \mid Z^{(0)}) \right].$$

*Then the (almost surely unique) minimizer of $\mathcal{L}_{\mathrm{tr}}(q)$ over $\mathcal{Q}$ is*

$$q^*(y \mid Z^{(0)}) = \sum_{c=1}^C \mathbf{1}_{\{y \in [\gamma_c, \gamma_{c+1})\}} \Delta_c^{-1} q_c^*(Z^{(0)}), \quad q_c^*(Z^{(0)}) := \int_{\gamma_c}^{\gamma_{c+1}} p_{(a,b]}(y \mid Z^{(0)})dy$$

*Equivalently, $q^*(\cdot \mid Z^{(0)})$ is obtained by bin-averaging $p(\cdot \mid Z^{(0)})$ on $(a, b]$ and then normalizing by the total mass $\int_a^b p(t \mid Z^{(0)})dt$, in addition to the $\Delta_c$ factor.*

*Proof.* Condition on $Z^{(0)}$. Any $q(\cdot \mid Z^{(0)}) \in \mathcal{Q}$ takes the form $q(y \mid Z^{(0)}) = \Delta_c^{-1} q_c(Z^{(0)})$ for $y \in [\gamma_c, \gamma_{c+1})$. Therefore,

$$\mathbb{E}_{y \sim p_{(a,b]}(\cdot \mid Z^{(0)})} \left[ -\log q(y \mid Z^{(0)}) \right] = \sum_{c=1}^C \int_{\gamma_c}^{\gamma_{c+1}} p_{(a,b]}(y \mid Z^{(0)}) \left[ -\log \left( \Delta_c^{-1} q_c(Z^{(0)}) \right) \right] dy$$

$$= \sum_{c=1}^C \left( -\log q_c(Z^{(0)}) + \log \Delta_c \right) \underbrace{\int_{\gamma_c}^{\gamma_{c+1}} p_{(a,b]}(y \mid Z^{(0)})dy}_{=:p_c(Z^{(0)})}.$$

The term $\sum_{c=1}^C (\log \Delta_c) p_c(Z^{(0)})$ does not depend on $q$, so minimizing the conditional expected NLL over $\mathcal{Q}$ is equivalent to minimizing $\sum_{c=1}^C p_c(Z^{(0)})(-\log q_c(Z^{(0)}))$ over the simplex $\{q_c \geq 0, \ \sum_c q_c = 1\}$. This is the categorical cross-entropy between $p(Z^{(0)}) = (p_c(Z^{(0)}))_{c=1}^C$ and $q(Z^{(0)}) = (q_c(Z^{(0)}))_{c=1}^C$, and its unique minimizer is $q_c(Z^{(0)}) = p_c(Z^{(0)})$ for all $c$. Thus, $q_c^*(Z^{(0)}) = p_c(Z^{(0)}) = \int_{\gamma_c}^{\gamma_{c+1}} p_{(a,b]}(y \mid Z^{(0)})dy$, which yields the stated $q^*$. Taking expectation over $Z^{(0)}$ proves the result. $\square$

## B.4. Kernel Ridge Regression (KRR)

**Theorem B.8** (Standard KRR). *Let $\{x_i, y_i\}_{1:n}$ with $x_i \in \mathcal{X}$, $y_i \in \mathbb{R}$, and let $\kappa : \mathcal{X} \times \mathcal{X} \to \mathbb{R}$ be a positive semidefinite kernel of RKHS $\mathcal{H}$ with norm $\|\cdot\|_{\mathcal{H}}$. Let $\sigma^2 > 0$ be the ridge parameter. Consider the kernel ridge regression (KRR) problem*

$$u^* \in \arg\min_{u \in \mathcal{H}} \sum_{i=1}^n (y_i - u(x_i))^2 + \sigma^2 \|u\|_{\mathcal{H}}^2.$$

*Let $G \in \mathbb{R}^{n \times n}$ where $G_{ij} = \kappa(x_i, x_j)$ be the kernel Gram matrix, and $\lambda_1(G)$, $\lambda_n(G)$ be its maximum and minimum eigenvalues. Suppose $\eta^{(l)} > 0$ satisfy $\sup_l \eta^{(l)} < 2/\{\lambda_1(G) + \sigma^2\}$, and $\sum_{l=0}^\infty \eta^{(l)} = \infty$. Define $u^{(0)} \equiv 0$ and*

$$u^{(l+1)}(\cdot) = (1 - \eta^{(l)} \sigma^2) u^{(l)}(\cdot) + \eta^{(l)} \sum_{i=1}^n \kappa(x_i, \cdot)(y_i - u^{(l)}(x_i)), \quad l \geq 0.$$

*Then for all $i \in [n]$, $u^{(l)}(x_i) \to u^*(x_i)$, and for any fixed $x \in \mathcal{X}$, $u^{(l)}(x) \to u^*(x)$. Furthermore, if $\eta^{(l)} = \eta \in \left(0, 1/(\lambda_1(G) + \sigma^2)\right)$ for all $l \geq 0$, then for $L \geq 1$,*

$$|u^{(L)}(x_i) - u^*(x_i)| \leq \exp\left(-(1-\rho)L\right) \frac{\lambda_1(G)}{\lambda_n(G) + \sigma^2} \|Y\|_2, \quad i \in [n],$$

$$|u^{(L)}(x) - u^*(x)| \leq \exp\left(-(1-\rho)L\right) \frac{\|k_x\|_2}{\lambda_n(G) + \sigma^2} \|Y\|_2.$$

*where $\rho := 1 - \eta(\lambda_n(G) + \sigma^2)$ is the convergence factor.*

*Proof.* By the representer theorem, we know that $u^*(\cdot) = \sum_{i=1}^n \alpha_i \kappa(x_i, \cdot)$ for some $\alpha = [\alpha_1, \cdots, \alpha_n]^\top \in \mathbb{R}^n$. Therefore, we can write $u_X^* := [u^*(x_1), \cdots, u^*(x_n)]^\top = G\alpha$, and note that $\|u^*\|_{\mathcal{H}}^2 = \langle \sum_i \alpha_i \kappa(x_i, \cdot), \sum_i \alpha_j \kappa(x_j, \cdot) \rangle_{\mathcal{H}} = \sum_{ij} \alpha_i \alpha_j \kappa(x_i, x_j) = \alpha^\top G \alpha$. Combined, the KRR is equivalent to a quadratic optimization

$$\min_{\alpha \in \mathbb{R}^n} \|Y - G\alpha\|_2^2 + \sigma^2 \alpha^\top G \alpha$$

where $Y := [y_1, \cdots, y_n]^\top$. Solving the gradient $-2G(Y - G\alpha) + 2\sigma^2 G\alpha = 0$, the minimum satisfies $G(G + \sigma^2 I_n)\alpha = GY$, and after rearranging, we have

$$u_X^* = G\alpha = (G + \sigma^2 I_n)^{-1} GY = G(G + \sigma^2 I_n)^{-1} Y \tag{16}$$

(since if two invertible matrices $A$ and $B$ commute, so do $A$ and $B^{-1}$).

On the other hand, for a test input $x \in \mathcal{X}$, its function value is $u^*(x) = \sum_i \alpha_i \kappa(x_i, x) = k_x^\top \alpha$, where $(k_x)_i = \kappa(x_i, x)$ for $i \in [n]$. We know from (16) that $\alpha = (G + \sigma^2 I_n)^{-1} Y + \delta$ where $\delta \in \ker(G)$; hence $k_x^\top \alpha = k_x^\top (G + \sigma^2 I_n)^{-1} Y + k_x^\top \delta$. Let $f_\delta(\cdot) := \sum_{i=1}^n \delta_i \kappa(x_i, \cdot) \in \mathcal{H}$ be the function in RKHS defined by $\delta$. Since $\delta \in \ker(G)$, we have $\|f_\delta\|_{\mathcal{H}}^2 = \delta^\top G \delta = 0$, which implies that for any $x \in \mathcal{X}$, $f_\delta(x) = \langle f_\delta(\cdot), \kappa(x, \cdot) \rangle_{\mathcal{H}} = \sum_{i=1}^n \delta_i \kappa(x_i, x) = k_x^\top \delta = 0$. Therefore,

$$u^*(x) = k_x^\top \alpha = k_x^\top (G + \sigma^2 I_n)^{-1} Y. \tag{17}$$

Note that $u_X^*$ is a solution to the linear equation $(G + \sigma^2 I_n) u_X^* = GY$. Define the corresponding Richardson iteration with $\{\eta^{(l)}\}_{l \geq 0}$ as the *training recursion*:

$$u_X^{(l+1)} = (1 - \eta^{(l)} \sigma^2) u_X^{(l)} + \eta^{(l)} G(Y - u_X^{(l)}). \tag{18}$$

Equivalently, for each coordinate $j \in [n]$,

$$u^{(l+1)}(x_j) = (1 - \eta^{(l)} \sigma^2) u^{(l)}(x_j) + \eta^{(l)} \sum_{i=1}^n k(x_i, x_j)(y_i - u^{(l)}(x_i)) \tag{19}$$

**Convergence of the training recursion.** We repeat the same argument given in Section A. Rearranging (18) in terms of the error $u_X^{(l)} - u_X^*$, and noting that $(G + \sigma^2 I_n) u_X^* = GY$, we have

$$u_X^{(l+1)} - u_X^* = \left(I_n - \eta^{(l)}(G + \sigma^2 I_n)\right)(u_X^{(l)} - u_X^*)$$

Using the spectral decomposition $G = Q\Lambda Q^\top$, we have $I_n - \eta^{(l)}(G + \sigma^2 I_n) = Q(I_n - \eta^{(l)}(\Lambda + \sigma^2 I_n))Q^\top$; hence

$$\|u_X^{(l+1)} - u_X^*\|_2 \leq \left(\max_i \left|1 - \eta^{(l)}(\lambda_i(G) + \sigma^2)\right|\right) \|u_X^{(l)} - u_X^*\|_2.$$

Therefore, (18) converges if $0 < \eta^{(l)}(\lambda_1(G) + \sigma^2) < 2$ for all $l \geq 0$, $\sup_l \eta^{(l)} < 2/\{\lambda_1(G) + \sigma^2\}$, and $\sum_{l=0}^\infty \eta^{(l)} = \infty$.

**Convergence rate of the training recursion.** Suppose $\eta^{(l)} = \eta \in \left(0, (\lambda_1(G) + \sigma^2)^{-1}\right)$ is a constant. Then we have $\max_i \left|1 - \eta(\lambda_i(G) + \sigma^2)\right| = 1 - \eta(\lambda_n(G) + \sigma^2)$, and noting that $u^{(0)} := 0$,

$$\begin{aligned}
\|u_X^{(L)} - u_X^*\|_2 &\leq \left(1 - \eta\left(\lambda_n(G) + \sigma^2\right)\right)^L \|u_X^*\|_2 \\
&\leq \exp\left(-\eta(\lambda_n(G) + \sigma^2)L\right) \|G(G + \sigma^2 I_n)^{-1} Y\|_2 \\
&\leq \exp\left(-\eta(\lambda_n(G) + \sigma^2)L\right) \frac{\lambda_1(G)}{\lambda_n(G) + \sigma^2} \|Y\|_2.
\end{aligned}$$

**Convergence of the testing recursion.** For each $u_X^{(l)}$, define the *testing recursion* $u^{(l)}(x)$ as $u^{(0)}(x) := 0$ and for $l \geq 0$,

$$u^{(l+1)}(x) = (1 - \eta^{(l)}\sigma^2)u^{(l)}(x) + \eta^{(l)}\sum_{i=1}^{n}k(x_i, x)(y_i - u^{(l)}(x_i))$$
$$= (1 - \eta^{(l)}\sigma^2)u^{(l)}(x) + \eta^{(l)}k_x^\top(Y - u_X^{(l)}). \tag{20}$$

If $0 < \eta^{(l)}(\lambda_1(G) + \sigma^2) < 2$ and $\sum_{l=0}^{\infty}\eta^{(l)} = \infty$, we know that $u^{(l)} \to u^*$, and we can see that (20) still holds if we plug $u_X^*$ into $u_X^{(l)}$ and $u^*(x)$ into $u^{(l+1)}(x)$ and $u^{(l)}(x)$:

$$u^*(x) = (1 - \eta^{(l)}\sigma^2)u^*(x) + \eta^{(l)}k_x^\top(Y - u_X^*). \tag{21}$$

Subtracting (21) from (20) and writing $e_X^{(l)} := u_X^{(l)} - u_X^*$ and $e^{(l)}(x) := u^{(l)}(x) - u^*(x)$, we have

$$e^{(l+1)}(x) = (1 - \eta^{(l)}\sigma^2)e^{(l)}(x) - \eta^{(l)}k_x^\top e_X^{(l)}. \tag{22}$$

The first term is a contraction since $|1 - \eta^{(l)}\sigma^2| < 1$ under our assumption, and the second term vanishes as $u^*$ converges. Therefore, $e^{(l)}(x) \to 0$, i.e., $u^{(l)}(x) \to u^*(x)$.

**Convergence rate of the testing recursion.** Since $u_X^{(0)} := 0$ and $u^{(0)}(x) := 0$, we can write

$$e_X^{(0)} = -u_X^* = -G(G + \sigma^2 I_n)^{-1}Y, \quad e^{(0)}(x) = -u^*(x) = -k_x^\top(G + \sigma^2 I_n)^{-1}Y.$$

Suppose $\eta^{(l)} = \eta \in (0, (\lambda_1(G) + \sigma^2)^{-1})$. Define $a := 1 - \eta\sigma^2$, $A := I - \eta(G + \sigma^2 I_n) = aI_n - \eta G$. Since the training error proceeds as $e_X^{(l)} = A^l e_X^{(0)}$, accumulating (22), we see that the test recursion error rolls out as

$$e^{(l)}(x) = a^l e^{(0)}(x) - \eta k_x^\top \left(a^{l-1}I_n + a^{l-2}A + \cdots + A^{l-1}\right)e_X^{(0)}$$
$$= \left(-a^l k_x^\top + k_x^\top \left(\sum_{j=0}^{l-1}a^{l-1-j}A^j\right)\eta G\right)(G + \sigma^2 I_n)^{-1}Y.$$

Note that $\eta G = aI_n - A$ by the definition, and

$$\left(\sum_{j=0}^{l-1}a^{l-1-j}A^j\right)\eta G = \left(\sum_{j=0}^{l-1}a^{l-1-j}A^j\right)(aI_n - A) = a^l I_n - A^l.$$

Therefore, $e^{(l)}(x) = -k_x^\top \left(I - \eta(G + \sigma^2 I_n)\right)^l (G + \sigma^2 I_n)^{-1}Y$. Taking absolute value, we see that for $L \geq 0$,

$$|u^{(L)}(x) - u^*(x)| \leq \exp\left(-\eta(\lambda_n(G) + \sigma^2)L\right)\frac{\|k_x\|_2}{\lambda_n(G) + \sigma^2}\|Y\|_2.$$

which completes the proof. $\square$

## C. Proof of Main Theorems

Theorem 5.1 follows from Lemma B.3-B.4, and Theorem 5.2 is a restatement of Lemma B.6.

## C.1. Proof of Theorem 3.1

*Proof.* Set token dimension $d' = d + 4$. Let $\vartheta$ be (unspecified elements are 0), for $l = 0, \cdots, L - 1$,

$$K^{(l)} = Q^{(l)} = \text{diag}((1_d^\top, 0_4^\top)),$$
$$V_{d+2,d+1}^{(0)} = 1, \quad V_{d+3,d+1}^{(l)} = V_{d+4,d+2}^{(l)} = \eta^{(l)}, \quad V_{d+3,d+3}^{(l)} = V_{d+4,d+4}^{(l)} = -\eta^{(l)},$$
$$S_{d+3,d+3}^{(l)} = S_{d+4,d+4}^{(l)} = -\eta^{(l)}\sigma^2, \tag{23}$$
$$W = \begin{bmatrix} 0_d^\top & 0 & 0 & 1 & 0 \\ 0_d^\top & \sigma^2 & 1 & 0 & -1 \end{bmatrix} \in \mathbb{R}^{2\times(d+4)}.$$

where $\eta^{(l)} > 0$ are chosen such that $\sup_l \eta^{(l)} < 2/\{\lambda_1(G) + \sigma^2\}$ and $\sum_{l=0}^\infty \eta^{(l)} = \infty$.

**Step 1 (First attention layer output).** For $l = 0, \cdots, L$, we write $K^{(l)}Z^{(l)} = [k_1^{(l)}, \cdots, k_{n+1}^{(l)}]$, $Q^{(l)}Z^{(l)} = [q_1^{(l)}, \cdots, q_{n+1}^{(l)}]$, and $V^{(l)}Z^{(l)} = [v_1^{(l)}, \cdots, v_{n+1}^{(l)}]$. The first attention layer in (9) is expressed as, for $j = 1, \ldots, n+1$,

$$[\text{Attn}_{M^{(0)},\kappa}(Z^{(0)}; V^{(0)}, K^{(0)}, Q^{(0)})]_{:,j} = \sum_{i=1}^{n+1} \kappa(k_i^{(0)}, q_j^{(0)}) M_{ij}^{(0)} v_i^{(0)}$$
$$= \kappa(k_{n+1}^{(0)}, q_j^{(0)}) v_{n+1}^{(0)}$$
$$= [0_{d+1}^\top, \kappa(x_{n+1}, x_j), 0, 0]^\top.$$

where second equality follows from $M_{i,j}^{(0)} = 1_{\{i>n\}}$ and third equality follows from $v_{n+1}^{(0)} = (0_{d+1}^\top, 1, 0, 0)^\top$ (due to $V_{d+2,d+1}^{(0)} = 1$) and $K^{(0)} = Q^{(0)} = \text{diag}((1_d^\top, 0_4^\top))$. Therefore,

$$Z^{(1)} = \begin{bmatrix} X^\top & x_{n+1} \\ Y^\top & 1 \\ k_x^\top & \kappa(x_{n+1}, x_{n+1}) \\ 0_{2\times n} & 0_2 \end{bmatrix}$$

**Step 2 (Richardson iteration implementation)**

For $l \geq 1$, (10) gives us, for $j = 1, \ldots, n+1$,

$$[Z^{(l+1)}]_{:,j} = [Z^{(l)}]_{:,j} + \sum_{i=1}^{n+1} \kappa(k_i^{(l)}, q_j^{(l)}) M_{ij} v_i^{(l)} + S^{(l)}[Z^{(l)}]_{:,j}$$
$$= [Z^{(l)}]_{:,j} + \eta^{(l)} \sum_{i=1}^{n} \kappa(x_i, x_j) \begin{bmatrix} 0_{d+2} \\ y_i - [Z^{(l)}]_{d+3,i} \\ \kappa(x_{n+1}, x_i) - [Z^{(l)}]_{d+4,i} \end{bmatrix} - \eta^{(l)}\sigma^2 \begin{bmatrix} 0_{d+2} \\ [Z^{(l)}]_{d+3,j} \\ [Z^{(l)}]_{d+4,j} \end{bmatrix}$$

where recall that $M_{i,j} = 1_{\{i\leq n\}}$. The last two rows of $Z^{(l)}$, denoted as $[Z^{(l)}]_{d+3,:}$ and $[Z^{(l)}]_{d+4,:}$, correspond to the following recursions: initiating from $f^{(0)}(\cdot) \equiv 0$ and $g^{(0)}(\cdot) \equiv 0$,

$$f^{(l+1)}(x_j) = (1 - \eta^{(l)}\sigma^2) f^{(l)}(x_j) + \eta^{(l)} \sum_{i=1}^{n} \kappa(x_i, x_j) \left( y_i - f^{(l)}(x_i) \right) \tag{24}$$

$$g^{(l+1)}(x_j) = (1 - \eta^{(l)}\sigma^2) g^{(l)}(x_j) + \eta^{(l)} \sum_{i=1}^{n} \kappa(x_i, x_j) \left( (k_x)_i - g^{(l)}(x_i) \right) \tag{25}$$

Assuming $\eta^{(l)} \in \left(0, 2/(\lambda_1(G) + \sigma^2)\right)$ for all $l \geq 1$ and $\sum_{l=0}^\infty \eta^{(l)} = \infty$, as $l \to \infty$, by Theorem B.8,

$$[f^{(l)}(x_1), \cdots, f^{(l)}(x_n)]^\top \to K(G + \sigma^2 I_n)^{-1} Y,$$
$$f^{(l)}(x_{n+1}) \to k_x^\top (G + \sigma^2 I_n)^{-1} Y,$$
$$[g^{(l)}(x_1), \cdots, g^{(l)}(x_n)]^\top \to K(G + \sigma^2 I_n)^{-1} k_x,$$
$$g^{(l)}(x_{n+1}) \to k_x^\top (G + \sigma^2 I_n)^{-1} k_x.$$

Therefore, for any column $j \in [n+1]$, the rows $[Z^{(l+1)}]_{d+3:,j}$ and $[Z^{(l+1)}]_{d+4:,j}$ evolve as (24) and (25) respectively, each starting from 0. For large enough $L$, $[Z^{(L)}]_{d+3:,n+1} \approx \mu(Z^{(0)})$, and $\sigma^2 + [Z^{(l+1)}]_{d+2:,n+1} - [Z^{(l+1)}]_{d+4:,n+1} \approx \tau(Z^{(0)})$. The convergence rates follow from Theorem B.8. $\qquad\square$

### C.2. Proof of Theorem 4.1

*Proof.* We first state the regularity conditions. For GP regression, the posterior predictive distribution is Gaussian with $\tau(Z^{(0)}) \geq \sigma^2 > 0$, so these conditions hold on any fixed truncation interval as long as the predictive mean and variance remain in a compact subset. We assume, uniformly over the context $Z^{(0)}$,

1. $\left(\mu(Z^{(0)}), \tau(Z^{(0)})\right) \in \mathcal{K}$ for some $\mathcal{K} \subset \mathbb{R} \times (0,\infty)$ and $\psi$ is approximated uniformly on $\mathcal{K}$ by MLP $\tilde{\psi}$ with a non-polynomial activation,

2. the target PPD belongs to a one-dimensional exponential family $f_\theta(y)$,

3. $f_\theta(y)$ is Lipschitz continuous on $(a, b]$ and the grid is equidistant.

Fix a compact set $\mathcal{K} \subset \mathbb{R} \times (0,\infty)$ on which the mapping $\psi : \mathcal{K} \to \mathbb{R}^2$ is defined as $\psi : (\mu, \tau) \mapsto \left(\frac{\mu}{\tau}, -\frac{1}{2\tau}\right)$. Such $\mathcal{K}$ is plausible given that $y \in (a, b]$ and $\tau(Z^{(0)}) \in [\sigma^2, \sigma^2 + \kappa(x, x)]$ for any $Z^{(0)}$. By the universal approximation theorem (Hornik, 1991; Leshno et al., 1993), for any $\delta_{\text{MLP}} > 0$, there exists a single layer MLP $\tilde{\psi} : \mathcal{K} \to \mathbb{R}^2$ of the form

$$\tilde{\psi}(\theta) = \tilde{W}_2 \, \text{act}(\tilde{W}_1 \theta + \tilde{h}_1) + \tilde{h}_2,$$

with a non-polynomial activation function act (e.g., ReLU), such that $\sup_{\theta \in \mathcal{K}} \|\psi(\theta) - \tilde{\psi}(\theta)\|_2 \leq \delta_{\text{MLP}}$. For a Lipschitz activation, $\tilde{\psi}$ is also Lipschitz; for any $x, x' \in \mathcal{K}$, $\|\tilde{\psi}(x) - \tilde{\psi}(x')\|_2 \leq L_{\tilde{\psi}} \|x - x'\|_2$.

Define the midpoint $\xi_c := (\gamma_c + \gamma_{c+1})/2$ and $\Xi \in \mathbb{R}^{C \times 2}$ as $\Xi_{c,:} = [\xi_c, \xi_c^2]^\top$ for $c \in [C]$. Fix $Z^{(0)}$. Let $\text{TF}_{L,\theta}(Z^{(0)}) =: \left[\mu_L(Z^{(0)}), \tau_L(Z^{(0)})\right]^\top$. We suppress the notation $Z^{(0)}$ and define the logits $\ell_1, \ell_2, \ell_3 \in \mathbb{R}^C$ as

$$\ell_1 := \Xi\tilde{\psi}(\mu_L, \tau_L) \quad \text{Transformer logit,}$$
$$\ell_2 := \Xi\tilde{\psi}(\mu, \tau) \quad \text{exact moments } \mu, \tau,$$
$$\ell_3 := \Xi\psi(\mu, \tau) \quad \text{exact natural parameter conversion } \psi.$$

Recall that by Theorem 3.1, $\|(\mu, \tau) - (\mu_L, \tau_L)\|_\infty \lesssim \exp(-(1-\rho)L)$ which implies $\|(\mu, \tau) - (\mu_L, \tau_L)\|_2 \lesssim \exp(-(1-\rho)L)$. Therefore,

$$\|\ell_1 - \ell_2\|_2 \leq \|\Xi\|_2 \left\|\tilde{\psi}(\mu, \tau) - \tilde{\psi}(\mu_L, \tau_L)\right\|_2 \leq \|\Xi\|_2 L_{\tilde{\psi}} \|(\mu, \tau) - (\mu_L, \tau_L)\|_2 \lesssim \exp(-(1-\rho)L),$$

where $\|\Xi\|_2$ is the fixed, largest singular value of $\Xi$. Moreover, by the definition of $\tilde{\psi}$, it holds that $\|\ell_2 - \ell_3\|_2 \leq \|\Xi\|_2 \delta_{\text{MLP}}$. Combined, we have

$$\|\ell_1 - \ell_3\|_\infty \leq \|\ell_1 - \ell_3\|_2 \lesssim \delta_{\text{MLP}} + \exp(-(1-\rho)L).$$

Define probability vectors $p_l := \text{sm}(\ell_l)$ for $l \in \{1, 3\}$, and write $\Delta$ as an upper bound of the $\ell_\infty$ difference of the logits $\|\ell_1 - \ell_3\|_\infty \leq \Delta$. This implies that, for any $c \in [C]$,

- $(\ell_1)_c - (\ell_3)_c \in [-\Delta, \Delta]$

- $e^{(\ell_1)_c} = e^{(\ell_3)_c} e^{(\ell_1)_c - (\ell_3)_c} \in [e^{-\Delta} e^{(\ell_3)_c}, e^\Delta e^{(\ell_3)_c}]$

- $\sum_c e^{(\ell_1)_c} \in [e^{-\Delta} \sum_c e^{(\ell_3)_c}, e^\Delta \sum_c e^{(\ell_3)_c}]$

Therefore, $\frac{(p_1)_c}{(p_3)_c} = e^{(\ell_1)_c - (\ell_3)_c} \frac{\sum_{c'} e^{(\ell_3)_{c'}}}{\sum_{c'} e^{(\ell_1)_{c'}}} \in [e^{-2\Delta}, e^{2\Delta}]$, and

$$(p_1)_c \leq e^{2\Delta}(p_3)_c \Rightarrow (p_1)_c - (p_3)_c \leq (e^{2\Delta} - 1)(p_3)_c,$$
$$(p_1)_c \geq e^{-2\Delta}(p_3)_c \Rightarrow (p_1)_c - (p_3)_c \geq (e^{-2\Delta} - 1)(p_3)_c.$$

Since $\Delta > 0$ and $(p_3)_c \leq 1$, this implies that, for a small $\Delta$,

$$\|p_1 - p_3\|_1 \leq e^{2\Delta} - 1 = O(\Delta)$$

We construct a piecewise continuous density on $(a, b]$ with probability vectors $p_1$ and $p_3$:

$$q_\vartheta(y \mid Z^{(0)}) := \sum_{c=1}^{C} \frac{\mathbf{1}_{y \in (\gamma_c, \gamma_{c+1}]}}{(b-a)/C} \, \mathrm{sm}\left(\Xi\tilde{\psi}(\mu_L, \tau_L)\right)_c,$$

$$q_3(y \mid Z^{(0)}) := \sum_{c=1}^{C} \frac{\mathbf{1}_{y \in (\gamma_c, \gamma_{c+1}]}}{(b-a)/C} \, \mathrm{sm}\left(\Xi\psi(\mu, \tau)\right)_c.$$

By Lemma B.1-B.2, $\|p_n - q_3\|_{L^1(a,b]} \lesssim 1/C + \varepsilon_{\mathrm{tail}}(Z^{(0)})$, and since $\|q_\vartheta - q_3\|_{L^1(a,b]} = \|p_1 - p_3\|_1$, we conclude that

$$\mathrm{TV}(p_n, q_\vartheta) \lesssim \exp(-(1-\rho)L) + \delta_{\mathrm{MLP}} + 1/C + \varepsilon_{\mathrm{tail}}(Z^{(0)}),$$

which completes the proof. $\qquad\qquad\square$

Although we focus on GP regression settings where PPD is Gaussian and $h$ is a constant, if the target exponential family distribution contains non-constant $h$ terms, the parameter mapping $\psi$ and data mapping $T$ can be appropriately expanded such that $h$ becomes a constant on its support. For example, if the target exponential density is an inverse Gaussian distribution parameterized by mean $\mu$ and variance $\tau$ so that $f_\theta(y) = \sqrt{\frac{\mu^3/\tau}{2\pi y^3}} \exp\left(-\frac{\mu(y-\mu)^2}{2y\tau}\right) \mathbf{1}_{\{y>0\}} = \exp\left(\langle(-0.5\mu/\tau, -0.5\mu^3/\tau), (y, 1/y)\rangle + \frac{\mu^2}{\tau} + \frac{1}{2}(\log\frac{\mu^3}{\tau} - \log(2\pi))\right) y^{-3/2} \mathbf{1}_{\{y>0\}}$, we parametrize it as $f_\theta(y) = \exp(\langle\psi(\mu, \tau), T(y)\rangle + \frac{\mu^2}{\tau} + \frac{1}{2}(\log\frac{\mu^3}{\tau} - \log(2\pi)))h(y)$ with $T(y) = (y, 1/y, \log y)$, $\psi(\mu, \tau) = (-\mu/(2\tau), -\mu^3/(2\tau), -3/2)$, and $h(y) = \mathbf{1}_{\{y>0\}}$.

### C.3. Proof of Theorem 5.3

*Proof.* Under the assumption, $\mathrm{TF}^{\mathrm{pr}}_{\vartheta,L}$ implements $L$ iterations of the preconditioned Richardson iteration solving (13). As such, the theorem addresses bounding the finite iteration convergence error of the two testing recursions to obtain the linear readouts $\mu(Z^{(0)})$ and the variance reduction term, which, from the proof of Theorem B.8, has the same convergence factor as the training recursion. Since the error is stated in terms of the maximum of the two, it suffices to show for one recursion, say, to obtain $\mu(Z^{(0)})$.

Without loss of generality, assume $\|\mu(Z^{(0)})\|_2 = 1$. Let $c_n := \mathrm{cond}(D^{-1}(G + \sigma^2 I_n))$. The convergence factor is, under the assumption of optimal step size, given by $\rho = 1 - 2/(c_n + 2)$ (see Section A). Let $u^{(L)} := \mathrm{TF}^{\mathrm{pr}}_{\vartheta,L}(Z^{(0)})$ and $u^* := \mu(Z^{(0)})$. Following Section A, suppose we have $\epsilon > 0$ such that

$$\|u^{(L)} - u^*\|_2 \leq \rho^L \leq \epsilon.$$

Rearranging and noting that $\rho \in (0, 1)$,

$$L > \frac{\log\epsilon}{\log\rho} = \frac{\log\epsilon}{\log\left(1 - \frac{2}{c_n+1}\right)} \geq \frac{\log\epsilon}{-\frac{2}{c_n+1}} = \frac{c_n + 1}{2}\log(1/\epsilon)$$

where we used the fact that $\log(1 - x) \leq -x$ around 0. The rest follows from the fact that $c_n = \Theta(n)$ almost surely for large $n$ (see Lemma B.6). $\qquad\qquad\square$

## D. Details of Experiments

**Default hyperparameter choice.** The effects of lengthscale $\ell$ and variance $\sigma^2$ are both-sided. For $\ell$, a smaller $\ell$ makes the GP rougher and more localized, making predictions harder, especially in higher $d$. However, it makes the associated linear system easier to solve. On the other hand, a larger $\ell$ makes the GP smoother but can worsen conditioning due to correlations. Larger $\sigma$ makes the prediction harder but improves the conditioning via regularization.

Throughout our experiments, we fix the noise variance at $\sigma^2 = 0.2$. For the BLR task with linear kernel, we use the diagonal covariance $\Sigma$ is segmented by input dimension $d$:

$$\Sigma_{k,k} = \begin{cases} 2.0 & \text{if } 0 \leq k < \lfloor \frac{d}{3} \rfloor \\ 1.0 & \text{if } \lfloor \frac{d}{3} \rfloor \leq k < \lfloor \frac{2d}{3} \rfloor \\ 0.4 & \text{if } \lfloor \frac{2d}{3} \rfloor \leq k < d \end{cases}$$

For the RBF task with RBF kernel, we set the amplitude at $\alpha = 1$ and bandwidth $\ell = 0.8$. Additionally, for *theory* mode for RBF task, we allow the step sizes of the drift term and the data residual terms (computed by the attention) to be different.

**MLP head.** To isolate the effect of the attention layers, we fix the conversion module $\psi(\mu, \tau)$ by hard-coding the analytic map from predicted moments $(\mu, \tau)$ to the natural parameters of a Gaussian density. In *learnable* mode, we additionally introduce two learned scalar rescaling $s_1$ and $s_2$ applied to the outputs of $\psi(\mu, \tau)$. We also fix the linear readout matrix $\Xi$ to its theoretical value.

**Pretraining details.** We optimize our models using the Adam optimizer with a base learning rate of $\text{LR}_{\text{base}}$ and no weight decay. We employ a cosine learning rate decay schedule with a linear warmup phase. The warmup period lasts for the first $5\%$ of the total training steps, after which the learning rate follows a cosine decay to a minimum value of $0.1 \times \text{LR}_{\text{base}}$. Gradients are clipped at a global norm of $1.0$ to ensure stability. The pretraining hyperparameters are summarized in Table 1. All the metrics in the figures are evaluated on 4096 evaluation samples.

| Task | Figure | $\text{LR}_{\text{base}}$ | $N$ |
|---|---|---|---|
| BLR (theory), train $\text{TF}_{\vartheta,L}$ | Fig. 2 | $2 \times 10^{-4}$ | $10^4$ |
| BLR (learnable), train $\text{TF}_{\vartheta,L}$ | Fig. 2 | $2 \times 10^{-4}$ | $2 \times 10^4$ |
| RBF (theory), train $\text{TF}_{\vartheta,L}$ | Fig. 2 | $1 \times 10^{-3}$ | $5 \times 10^4$ |
| RBF (learnable), train $\text{TF}_{\vartheta,L}$ | Fig. 2 | $2 \times 10^{-4}$ | $10^5$ |
| RBF (learnable), train $\text{TF}_{\vartheta,L}$ | Fig. 4 | $2 \times 10^{-4}$ | $10^5$ |
| RBF (learnable), train $\text{TF}^{\text{pr}}_{\vartheta,L}$ | Fig. 4 | $2 \times 10^{-4}$ | $10^5$ |
| RBF (learnable), train $\text{TF}^{\text{pr}}_{\vartheta,L}$ | Fig. 5-14 | $2 \times 10^{-4}$ | $10^5$ |

*Table 1.* Pretraining hyperparameters for BLR and RBF tasks. All runs use batch size 128 for $N$ iterations.

## E. Additional Experimental Results

**Sensitivity to lengthscale.** We conducted additional simulations to assess the sensitivity of our findings to the RBF lengthscale. We fix $\sigma^2 = 0.2$ as in Section 6.3, set $n_{\max} = 256$ and $d = 16$, and vary $\ell \in \{0.4, 0.8, 1.6\}$, including the main-paper setting $\ell = 0.8$ for comparison. In addition to interval coverage, we report the continuous ranked probability score (CRPS), a proper scoring rule for predictive cumulative distribution functions (Gneiting & Raftery, 2007); lower CRPS indicates better predictive distributions, although its absolute scale depends on the scale of the response. For each row, entries are ordered by attention depth $L \in \{8, 16, 32\}$. In this setting, CRPS decreases as $\ell$ increases, suggesting that

| $\ell$ | $n'$ | CRPS | 50% | 90% | 95% |
|---|---|---|---|---|---|
| 0.4 | 256 | .538, .547, .539 | .504, .487, .505 | .904, .899, .904 | .953, .946, .953 |
| | 1024 | .540, .540, .538 | .495, .485, .499 | .896, .887, .894 | .947, .943, .947 |
| 0.8 | 256 | .318, .310, .307 | .488, .499, .509 | .890, .901, .905 | .942, .951, .954 |
| | 1024 | .271, .262, .258 | .444, .478, .505 | .848, .877, .897 | .908, .932, .950 |
| 1.6 | 256 | .157, .151, .149 | .500, .498, .506 | .901, .900, .908 | .950, .948, .952 |
| | 1024 | .147, .137, .135 | .481, .479, .503 | .883, .884, .908 | .938, .943, .952 |

*Table 2.* Sensitivity to RBF lengthscale $\ell$ with $d = 16$, $\sigma^2 = 0.2$, $n_{\max} = 256$, and $L \in \{8, 16, 32\}$. Entries in each metric column are ordered by $L = 8, 16, 32$. CRPS denotes the continuous ranked probability score; lower is better.

smoother GP draws reduce the statistical difficulty of the prediction problem. Across all lengthscales, increasing depth generally improves CRPS, with the clearest improvement at $\ell = 0.8$. The coverage values remain reasonably close to nominal even at $n' = 1024 > n_{\max}$, especially for larger depths.

**Hierarchical GP prior.** We also evaluate a hierarchical GP setting with random hyperparameters. Let $\theta \sim \sum_{h=1}^{H} \pi_h \delta_{\theta_h}$ for $\theta_h = (\ell_h, \sigma_h)$. Conditional on $\theta = \theta_h$, $f \sim \mathrm{GP}(0, \kappa_{\ell_h})$, $y_i = f(x_i) + \epsilon_i$ where $\epsilon_i \overset{\text{iid}}{\sim} \mathcal{N}(0, \sigma_h^2)$. In this setting, the PPD is tractable as a finite mixture of Gaussian predictive distributions. For the experiment, we use the uniform prior over $\Theta = \{0.4, 0.8, 1.2\} \times \{0.1, 0.2, 0.3\}$. We set $n_{\max} = 256$, $d = 16$, and otherwise match the setup of Section 6.3. In each cell, entries are ordered by attention depth $L \in \{8, 16, 32\}$.

| $n'$ | CRPS | 50% | 90% | 95% |
|------|------|-----|-----|-----|
| 256  | .380, .378, .377 | .573, .575, .590 | .895, .897, .901 | .937, .937, .942 |
| 512  | .374, .371, .367 | .570, .568, .590 | .889, .891, .903 | .926, .925, .932 |
| 1024 | .358, .353, .346 | .560, .563, .598 | .887, .887, .902 | .930, .929, .940 |

*Table 3.* Hierarchical GP experiment with random hyperparameters $\theta = (\ell, \sigma) \in \{0.4, 0.8, 1.2\} \times \{0.1, 0.2, 0.3\}$, $d = 16$, $n_{\max} = 256$, and $L \in \{8, 16, 32\}$. Entries in each metric column are ordered by $L = 8, 16, 32$.

The results are consistent with the main findings. CRPS improves with depth $L$, while coverage remains reasonably calibrated for $n' > n_{\max}$, especially at the 90% and 95% levels. This suggests that the depth effect and normalized-attention mechanism identified in the fixed-hyperparameter GP setting remain relevant in this more complex hierarchical setting.

**Additional generalization results.** We repeat the generalization experiment from Figure 5 and 10 for dimensions $d \in \{4, 8\}$, with results shown in Figure 6-9. Across dimensions, the same pattern emerges: generalization improves with deeper $L$ and larger $n_{\max}$, except in the case $d = 8$, where the $L = 16$ model exhibits higher error than shallower counterparts, likely due to a training failure.

**Covariate shifts.** We replicate the generalization experiment under a covariate shift, where the evaluation inputs are drawn from $x_i \overset{\text{iid}}{\sim} \mathrm{Unif}[-1/\sqrt{d}, 1/\sqrt{d}]$ for $d \in \{4, 8\}$ The results shown in Figure 11-14. Despite the shift in covariates, we observe that the generalization performance is comparable to the true baseline for $n_{\max} = 512$ and $L \geq 16$, with $L = 16$ model at $d = 8$ being an exception.

# F. Details of Real Data Case Studies

We present a real data case study on the Sacramento home price dataset (Kuhn, 2008) and the Walker Lake dataset (Isaaks & Srivastava, 1989). Note that this case study is intended as an illustrative validation of the proposed mechanism rather than as a comprehensive benchmark.

**Sacramento.** The dataset is from the R package `caret` version 7.0.1 (Kuhn, 2008), from which we selected data corresponding to the cities of Sacramento and Elk Grove. We use the two spatial coordinates, longitude and latitude, as input features and denote the response by $V$. The spatial coordinates are centered and standardized coordinate-wise, and then divided by a constant 0.3; the response is centered and standardized with the scaling constant 1. This preprocessing is used consistently for empirical-Bayes fitting, PFN pretraining, and evaluation. Following Rasmussen & Williams (2005, Chapter 5), we first fit an empirical-Bayes anisotropic RBF GP to the Sacramento data. Specifically, we use the ARD kernel

$$\kappa(x, x') = \alpha^2 \exp\left(-\frac{1}{2}\sum_{k=1}^{2} \frac{(x_k - x'_k)^2}{\ell_k^2}\right),$$

together with Gaussian observation noise. The fitted hyperparameters are $\alpha = 0.666$, $\ell = (1.226, 0.770)$ and $\sigma = 0.836$. Using these hyperparameters, we pretrain the normalized transformer $\mathrm{TF}_{\vartheta, L}^{\mathrm{pr}}$ on synthetic data generated from the fitted prior. We use $C = 256$ bins, depth $L = 32$, and sample pretraining context sizes from $n' \sim \mathrm{Unif}[128, 552]$. The truncation interval $(a, b)$ for the binned output distribution is chosen by Monte Carlo calibration under the fitted prior, as in the synthetic experiments.

In Figure 1, we provide the trained PFN with the full Sacramento context and evaluate its PPD on a regular $100 \times 100$ spatial grid over the region $[-121.53, -121.33] \times [38.38, 38.69]$. From the PFN output probabilities, we compute the predictive

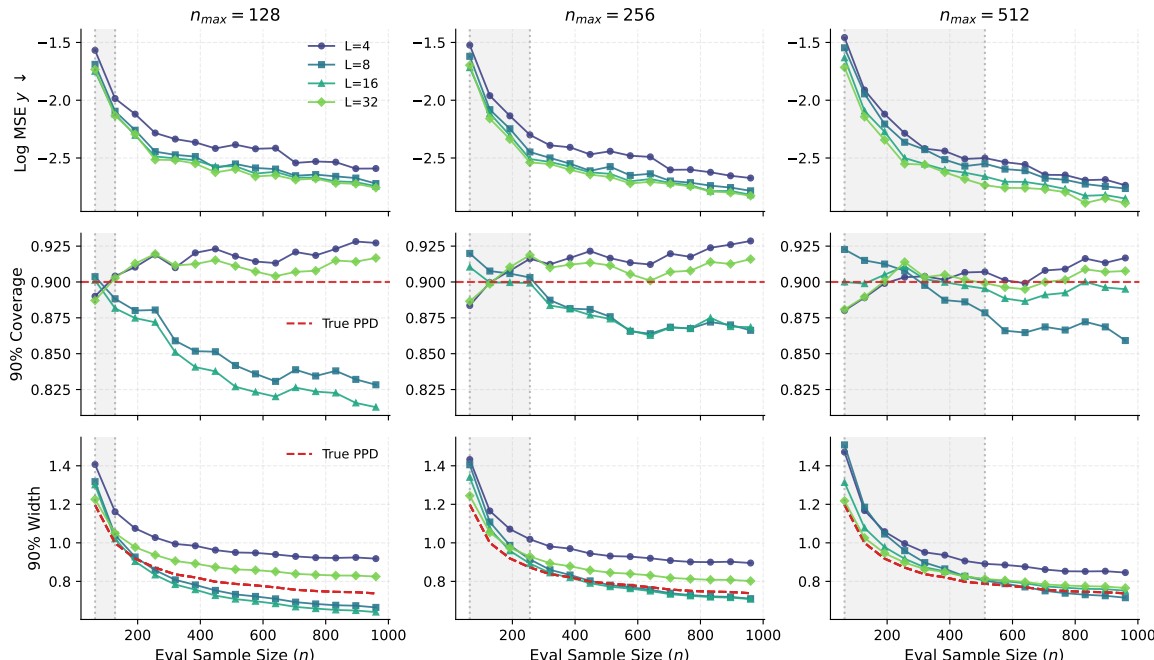

*Figure 6.* ($d = 4$) Prediction MSE, $90\%$ interval coverage, and $90\%$ interval width versus evaluation sample size for *learnable* normalized models with $C = 256$, depths $L \in \{4, 8, 16, 32\}$, and pretraining ranges $n \in [64, n_{\max}]$ with $n_{\max} \in \{128, 256, 512\}$. Red dashed curves denote the corresponding true PPD interval width/nominal coverage.

mean and the $5\%$ and $95\%$ predictive quantiles. As a baseline, we fit an empirical-Bayes GP to the same standardized context and evaluate its predictive mean and Gaussian predictive quantiles on the same grid. Finally, all predictions are transformed back to the original response scale.

**Walker Lake.** The dataset is from the R package `gstat` version 2.1-6 (Pebesma et al., 2015), whose spatial coordinates and response (V) are standardized in the same manner as Sacramento data. We estimate anisotropic RBF hyperparameters from a randomly sampled subset of the training data with $n = 200$ by maximizing the marginal log likelihood, obtaining $\alpha = 0.782$, $\ell_1 = 0.146$, $\ell_2 = 0.252$, and $\sigma = 0.611$. We then pretrain $\mathrm{TF}^{\mathrm{pr}}_{\vartheta, L}$ with $C = 256$ and $L = 32$ on synthetic data from the fitted prior with context sizes $n' \sim \mathrm{Unif}[128, 384]$. Given a separate context of size $n = 200$, the PFN PPD is evaluated on the full spatial grid and compared with an empirical-Bayes GP fitted to the same context. The resulting PFN and GP predictive means and $5\%/95\%$ quantiles show qualitatively similar spatial patterns (Figure 15).

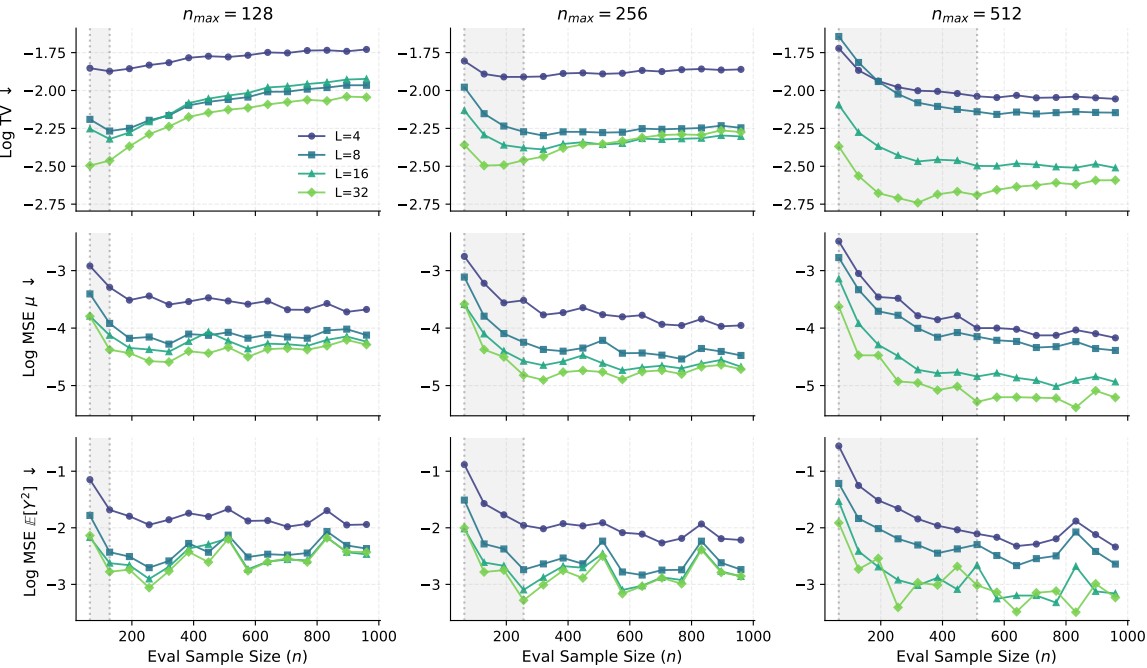

*Figure 7.* ($d = 4$) $\log \mathrm{E}[\mathrm{TV}(p_{(a,b]}, q_\vartheta)]$ and moment MSEs $\mathrm{E}(\mu - m_{\vartheta,1})^2$ and $\mathrm{E}(\tau + \mu^2 - m_{\vartheta,2})^2$ versus evaluation sample size $n'$ for *learnable* normalized models with $C = 256$, depths $L \in \{4, 8, 16, 32\}$, and pretraining ranges $n \in [64, n_{\max}]$ with $n_{\max} \in \{128, 256, 512\}$.

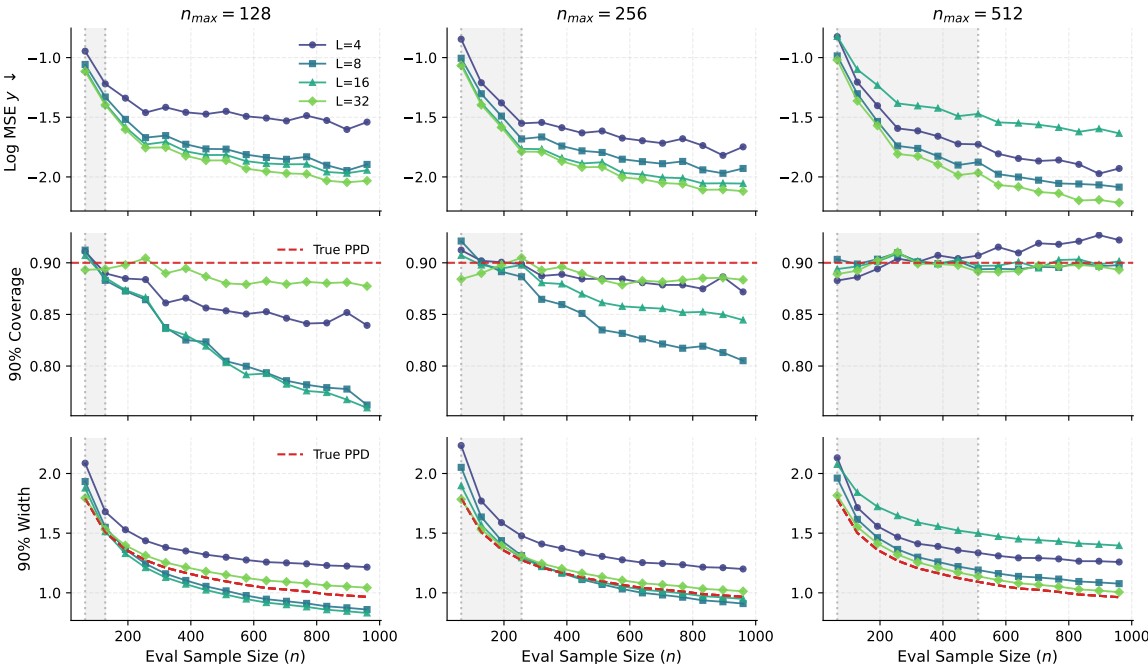

*Figure 8.* ($d = 8$) Prediction MSE, 90% interval coverage, and 90% interval width versus evaluation sample size for *learnable* normalized models with $C = 256$, depths $L \in \{4, 8, 16, 32\}$, and pretraining ranges $n \in [64, n_{\max}]$ with $n_{\max} \in \{128, 256, 512\}$. Red dashed curves denote the corresponding true PPD interval width/nominal coverage.

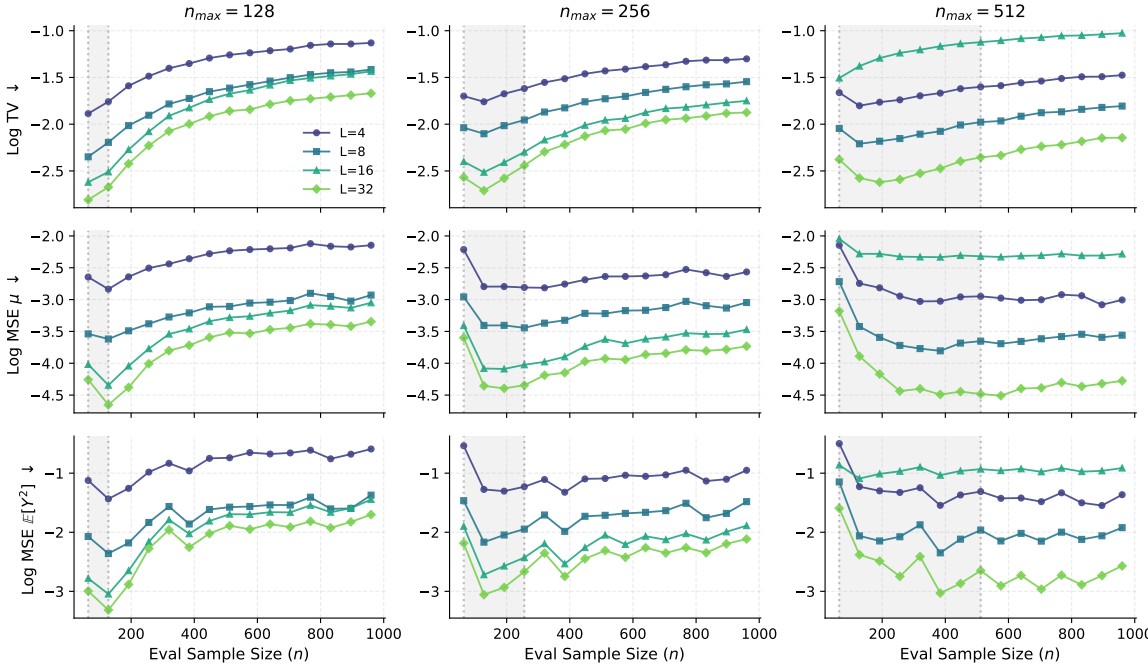

*Figure 9.* $(d = 8)$ $\log \mathrm{E}[\mathrm{TV}(p_{(a,b]}, q_\vartheta)]$ and moment MSEs $\mathrm{E}(\mu - m_{\vartheta,1})^2$ and $\mathrm{E}(\tau + \mu^2 - m_{\vartheta,2})^2$ versus evaluation sample size $n'$ for *learnable* normalized models with $C = 256$, depths $L \in \{4, 8, 16, 32\}$, and pretraining ranges $n \in [64, n_{\max}]$ with $n_{\max} \in \{128, 256, 512\}$.

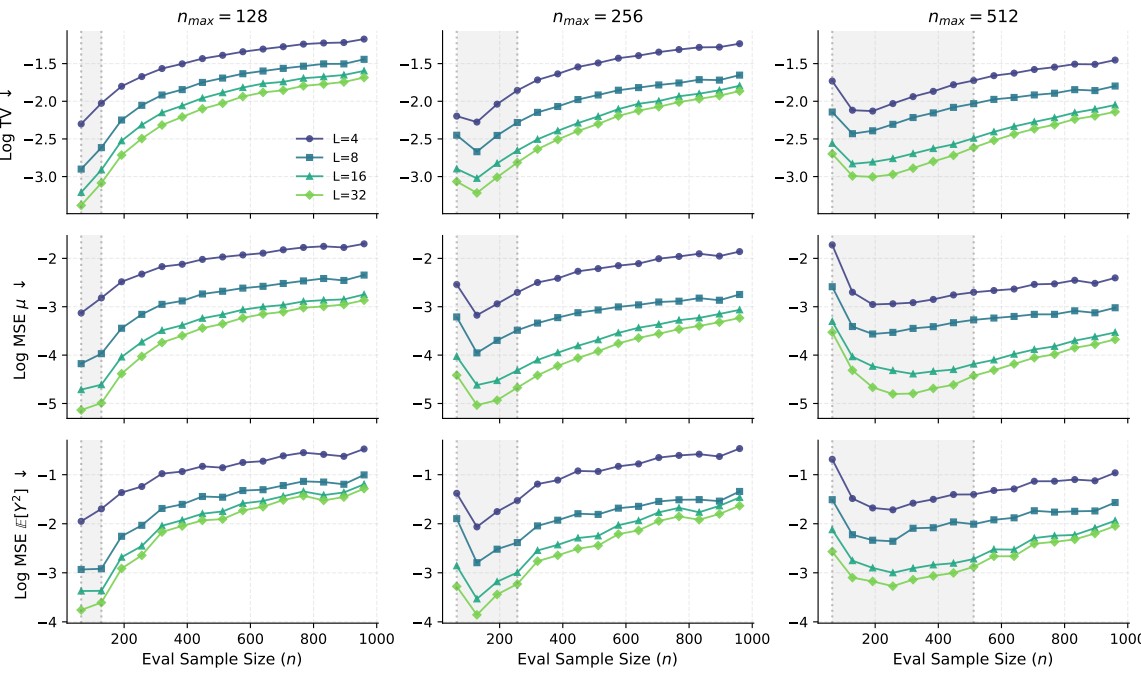

*Figure 10.* $(d = 16)$ $\log \mathrm{E}[\mathrm{TV}(p_{(a,b]}, q_\vartheta)]$ and moment MSEs $\mathrm{E}(\mu - m_{\vartheta,1})^2$ and $\mathrm{E}(\tau + \mu^2 - m_{\vartheta,2})^2$ versus evaluation sample size $n'$ for *learnable* normalized models with $C = 256$, depths $L \in \{4, 8, 16, 32\}$, and pretraining ranges $n \in [64, n_{\max}]$ with $n_{\max} \in \{128, 256, 512\}$. Errors decrease with larger $L$ and $n_{\max}$ and grow with $n'$, mirroring the trends in prediction and interval metrics in Figure 5. Consistent with Theorem 5.3, for fixed $L$ and $n_{\max}$, the error of the finite-iteration solver increases as the evaluation sample size grows.

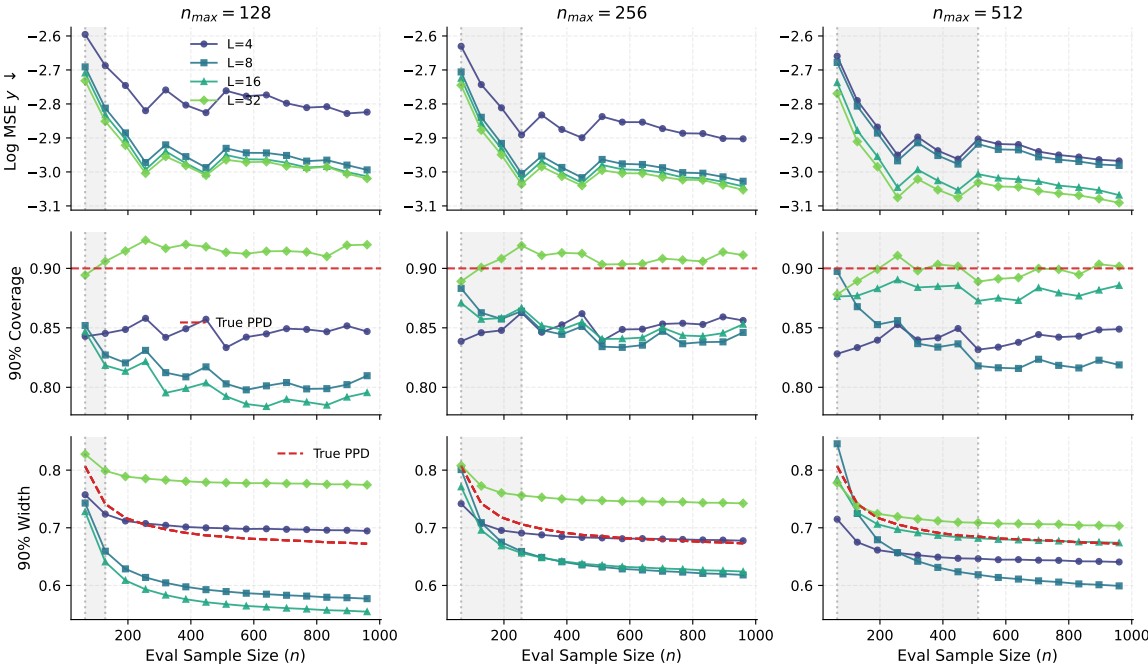

*Figure 11.* ($d = 4$, covariate shift) Prediction MSE, 90% interval coverage, and 90% interval width versus evaluation sample size for *learnable* normalized models with $C = 256$, depths $L \in \{4, 8, 16, 32\}$, and pretraining ranges $n \in [64, n_{\max}]$ with $n_{\max} \in \{128, 256, 512\}$. Red dashed curves denote the corresponding true PPD interval width/nominal coverage.

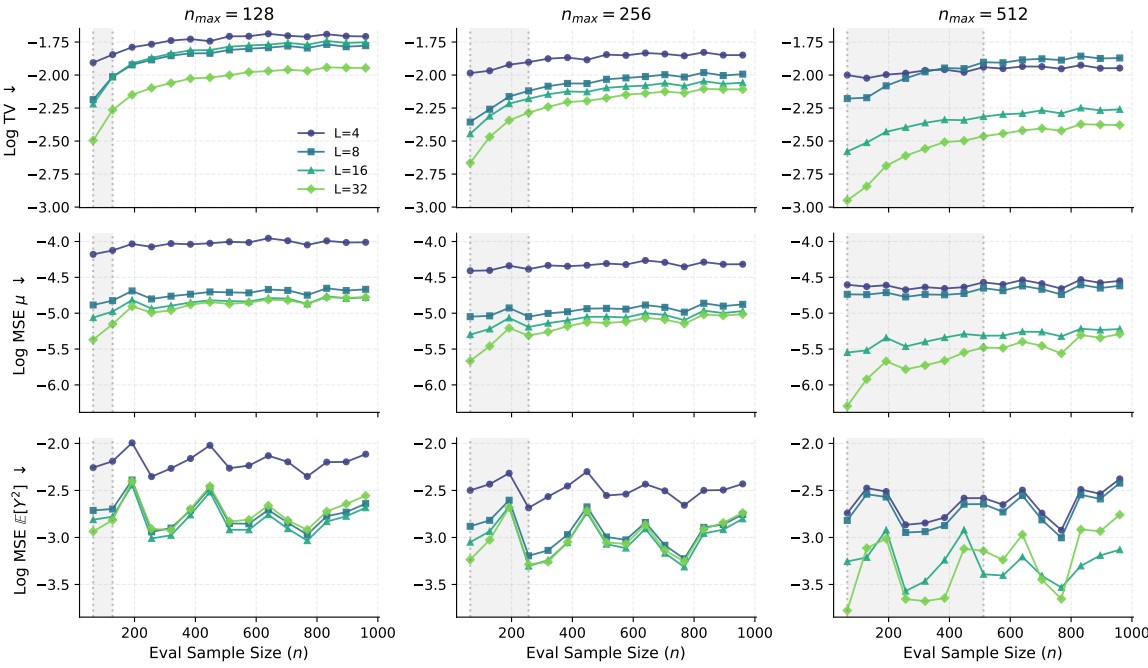

*Figure 12.* ($d = 4$, covariate shift) $\log \mathrm{E}[\mathrm{TV}(p_{(a,b)}, q_\vartheta)]$ and moment MSEs $\mathrm{E}(\mu - m_{\vartheta,1})^2$ and $\mathrm{E}(\tau + \mu^2 - m_{\vartheta,2})^2$ versus evaluation sample size $n'$ for *learnable* normalized models with $C = 256$, depths $L \in \{4, 8, 16, 32\}$, and pretraining ranges $n \in [64, n_{\max}]$ with $n_{\max} \in \{128, 256, 512\}$.

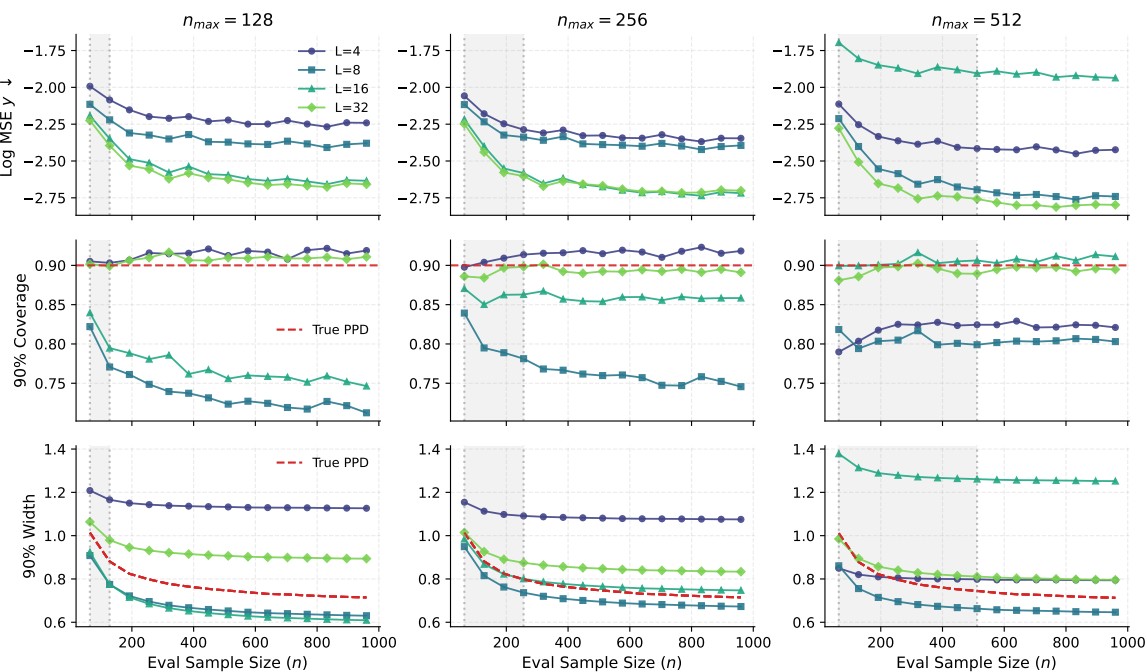

*Figure 13.* ($d = 8$, covariate shift) Prediction MSE, 90% interval coverage, and 90% interval width versus evaluation sample size for *learnable* normalized models with $C = 256$, depths $L \in \{4, 8, 16, 32\}$, and pretraining ranges $n \in [64, n_{\max}]$ with $n_{\max} \in \{128, 256, 512\}$. Red dashed curves denote the corresponding true PPD interval width/nominal coverage.

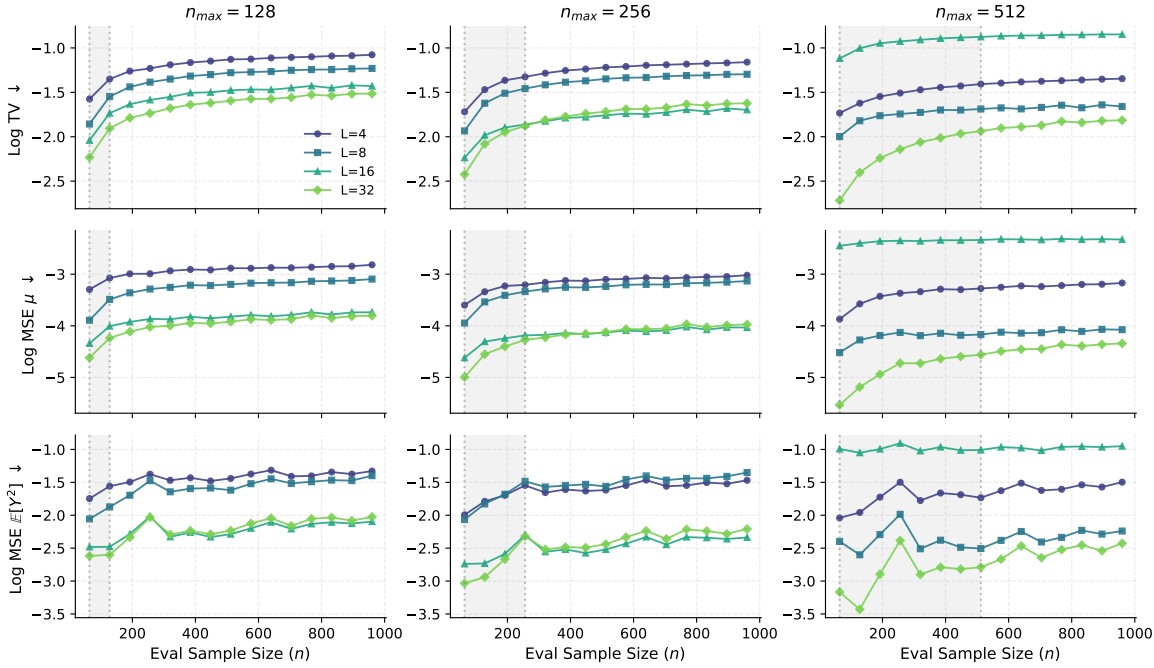

*Figure 14.* ($d = 8$, covariate shift) $\log \mathrm{E}[\mathrm{TV}(p_{(a,b]}, q_\vartheta)]$ and moment MSEs $\mathrm{E}(\mu - m_{\vartheta,1})^2$ and $\mathrm{E}(\tau + \mu^2 - m_{\vartheta,2})^2$ versus evaluation sample size $n'$ for *learnable* normalized models with $C = 256$, depths $L \in \{4, 8, 16, 32\}$, and pretraining ranges $n \in [64, n_{\max}]$ with $n_{\max} \in \{128, 256, 512\}$.

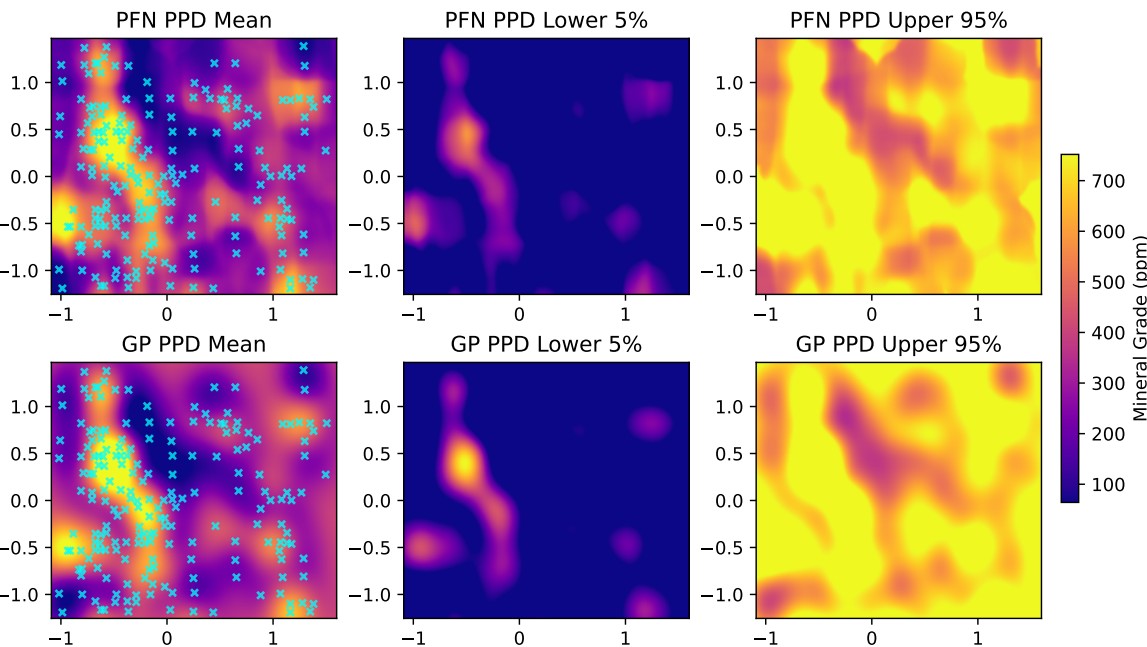

*Figure 15.* Illustration of similarities between PPDs produced by a transformer (PFN) and a Gaussian process (GP) on the Walker Lake dataset (Isaaks & Srivastava, 1989). The $x$, $y$ axes represent standardized spatial coordinates, while the color scale shows estimated mineral (V) concentrations in ppm. The top row displays PPD outputs from a transformer, while the bottom row shows PPD based on a GP. Columns correspond to the PPD mean (left), $5\%$ quantile (center), $95\%$ quantile (right) of PPD; both PPDs are based on the same 200 observations shown in blue $\times$ mark on the left column.

