# OpenReview forum: "Transformers Can Learn Posterior Predictive Distributions In-Context"
_ICML.cc/2026/Conference — ICML 2026 regular_

### Official Review · Reviewer_gA2c · 2026-02-23

**Soundness:** 3
**Presentation:** 3
**Significance:** 3
**Originality:** 3
**Overall Recommendation:** 5
**Confidence:** 3

**Summary:**

Theoretical understanding of PFN's ability to approximate Bayesian PPDs is lacking, despite their strong empirical performance. This is particularly true for in-context learning. The authors study GP regression, and show that transformers can approximate the mean and covariance of the PPD, in-context. This is an existence result, divorced from optimisation in the pretraining of the transformer. This result is extended to the case where an additional MLP layer of exponential family is used. The error bounds involve the bin resolution, depth and MLP approximation error. Empirical demonstrations of the theoretical results are given.

**Compliance With Llm Reviewing Policy:**

Affirmed.

**Final Justification:**

The rebuttal addressed my minor concerns. I still have concerns about the existence style result, but that is not unique to this work and I think this work is well-positioned in line with existing literature. All in all I find this work sound and to be of interest to the community. I therefore recommend acceptance.

**Key Questions For Authors:**

Please address my concerns above, referring to the more detailed points in the limitations/weaknesses in above. In particular:
- What are the precise mathematical conditions you require on the exponential family / MLP in order for your results to hold?
- Please contextualise the gap between existence / representation / approximation results and optimisation / pretraining. What can you say here to better convince the reader? And can you cite some more works?
- Can you clarify the paragraph following theorem 4.1?
- Please also address the more minor comments I listed above. If you run out of room in your rebuttal, I'm happy for you to ignore details of minor points to address the more important points.

**Limitations:**

The authors do not include a conclusion in the main paper, and I encourage the authors to include one in their final submission. This would give them a natural place to discuss limitations and future work, which seem to be largely absent at the moment.

Negative societal impact is not discussed, but I don't think they need to be particularly singled out for this largely theoretical work.

**Strengths And Weaknesses:**

**Strengths:**
- Theorem 3.1 shows deep (L layer) attention can represent GP mean and covariance to exponential precision in the depth, with a rate of convergence governed by some spectral properties of the prior kernel matrix evaluated on the data.
- Theorem 4.1 nicely extends to prediction of conditional expectations of discretised (scalar-valued) exponential family models (i.e. for supervised learning) by adding an MLP head to the previous model. It adds approximation error terms to theorem 1 related to the discretisation error and the MLP representation error.
- Both of these theorems nicely describe representation error of the transformer model in kernel ridge regression / GP settings (theorem 3.1) and beyond to conditional expectation of discretised exponential families. Both of these models consider *unnormalised attention*. The following results argue that normalisation is required in order to obtain in context generalisation, because it implements preconditioning.
- The topics considered are important current problems in understanding the representational ability of transformer models. Section 5 considers more or less properly normalised attention models, which is different to a lot of other investigations in this literature. As such, I consider this paper to be both *significant* and *original*.
- The paper is mostly *sound*. I have some isolated questions / issues, as discussed below.


**Weaknesses:**
- As is very common in this literature, results concern existence rather than actual learning (through pretraining of the transformer). For example, in Theorem 3.1, "there exists a set of parameters (given in 23) such that...". This kind of result seems to be present throughout this literature, and I'm not sure why such a style result is even valuable. Representation results are available for many models, and this doesn't seem to expose anything unique about transformers. This is probably me misunderstanding the literature rather than a true weakness of this particular work, because I have seen it many times. To make things worse, as is usual in this kind of result, the explicit identification of the weights in (23) have nothing to do with a model that one would expect to learn from something like gradient descent in pretraining the transformer. The explicit parameter values are really simple, but there must be a reason why practically gradient descent is actually used rather than these simple weights. I don't expect the authors to "fix" this weakness, but rather provide me and similar disheartened readers with some context for "there exists".
- I'm not sure if I follow the paragraph following theorem 4.1. I agree that when C is large and the probability that y falls outside of the discrete range is small, then the dominant term is the finite depth attention block. But when L is very large, and C is small, then the opposite is true. Are you equally saying that we could use a deep network without normalisation, and expect good representation error?


**Presentation / notation / terminology weaknesses:**
- "Attn" function is used in equation (4), before it is defined.
- The top row of figure 1 is labelled "TF", and the caption says "transformer (PFN)". Are these supposed to be the same label?
- As far as I can tell, $n$-generalisation is first mentioned on page 2, column 2. It seems to be defined as the ability of models to retain reliable point predictions and UQ beyond the pretraining sample size $n$. However, I am not sure if the $n$ in $n$-generalisation is the same as the sample size $n$ - it is never explicitly mentioned. It is mentioned that the dataset size is $n$ previously, but sometimes authors use different symbols to name concepts. It would be helpful to state what $n$ is close to the definition of $n$-generalisation.
- On page two it is stated that TabPFNv2 is pretrained on synthetic data up to $n=2048$, but it remains highly competitive for $n$ up to 10,000. Later on page 3, it is stated that PFN (I'm guessing, different to TabPFNv2?) is pretrained on synthetic data with varying sample sizes, in order to approximate the PPD for a brad range of $n$. This is confusing for two reasons: (1) the second statement only states a "broad range", not the explicit numbers (e.g. 100-10000). (2) The phrasing hints at a contradiction, in that the first statement seems to say that we don't need a broad range but the second statement seems to say that we do need to train on a broad range, but only because of lack of explanation.
- It is incorrectly stated near line 249 that $A(\theta)$ is the normalising constant of an exponential family, however it is the *log* normalising constant (log partition function).
- It is incorrectly stated that $h$ is the base measure, however the construct here necessarily implies $h$ is not a base measure but a base density.
- The authors do not include a conclusion in the main paper, and I encourage the authors to include one in their final submission. This would give them a natural place to discuss limitations and future work, which seem to be largely absent at the moment.

**Mathematical details / questions / soundness:**
- Checking Lemma B.2 in the appendix, it appears as though the exponential family model in question has to satisfy a finite supremum property (involving $\omega_p$), however this condition is not exposed in the main text.
- The (more precisely defined) exponential of the MLP $\widetilde{\psi}$ needs to be integrable with respect to the base density $h$, and this condition does not seem to be mentioned anywhere. We could have problems if e.g. $h \equiv 1$ and $\widetilde{\psi}$ is ReLU. More worrying is that at the bottom of column 1 page 5, the authors consider the case of constant $h$, which is valid for Gaussian because the sufficient statistics decay sufficiently fast when exponentiated, but would very often fail for common neural network activation functions.
- I believe the authors use a constant $h$ somewhere in their construction for theorem 4.1, or maybe the presentation is just ambiguous. In any case, the authors are very loose about exactly what conditions they require on the exponential family for Theorem 4.1 to hold.

---

> ### Author Rebuttal · Authors · 2026-03-31
>
> We thank the reviewer for the thoughtful feedback and address the main concerns below. In the revision, we will also incorporate the remaining comments and add a conclusion discussing limitations and future directions.
>
> > **[R4Q1]**
> > ... some context for "there exists".
> > Please contextualise the gap between existence / representation / approximation results and optimisation / pretraining. ... cite some more works?
>
> **[R4A1]** We appreciate this concern and agree that existence results can feel unsatisfying. We will revise the paper to distinguish:
> 1. can the architecture represent the target computation (existence);
> 2. how well can a model approximate it (approximation);
> 3. will pretraining find such a mechanism (pretraining).
>
> Our paper addresses mainly 1 and partly 2, but not 3.
>
> The main value of existence result, in our view, is **interpretability**. Pretrained transformers are hard to interpret, and it is often unclear whether the architecture can even realize the target (PPD) computation at all. An explicit construction addresses this first question. Such results are especially useful with *simple and explicit* construction as ours: attention layers *exactly* implement a recursion for predictive moments, while a shallow MLP maps those summaries to distribution bins, yielding a decomposable mechanism for PPD computation.
>
> Moreover, constructive results can motivate flexible but still interpretable model classes that help explain practical performance. As the reviewer noted, our proof uses simple weights for tractability. But our learnable transformer is a structured relaxation: the $Q,K$ maps remain diagonal and the sparsity pattern is preserved, so it can still be interpreted as Richardson-style iteration, now with layerwise-varying geometry and step sizes. This flexibility allows a fixed transformer to handle KRR problems across different context lengths. Related work similarly interprets such relaxations as preconditioned GD (Ahn et al., 2023) or anisotropic denoising (Rosu et al., 2025).
>
> That said, we fully agree that proving optimization guarantees beyond simplified settings remains an important future direction (Zhang et al., 2024), and our work is only a first step toward that goal. We will revise the paper to make this scope explicit and add the relevant references.
>
>
> > **[R4Q2]** Can you clarify the paragraph following thrm 4.1?
>
> **[R4A2]** We agree that Thrm 4.1 deserves clearer discussion. Its contribution lies less in the proof than in relating approximation accuracy to architectural choices, including depth $L$, MLP width, the number of bins $C$, and the truncation interval.
>
> We also agree that depth $L$ is not always the dominant source of error. In practice, however, the attention stack is often the main architectural component of interest, whereas the truncation interval, $C$, and MLP width are typically chosen large enough. In that regime, the theorem helps explain why increasing depth is the most natural route to improving accuracy.
>
> > **[R4Q3]** I believe the authors use a constant $h$ ...
>
> **[R4A3]** Constant $h$ is assumed throughout the proof, which we will make it explict. However, if $h$ is non-constant, we could consider an expanded notions of $\psi$ and $T$ that includes an additional component so that $h$ becomes constant over its support. For example, inverse Gaussian parametrized by mean $\mu$ and variance $\tau$ admit 2 representations:
>
>  * $T(y)=(y,1/y)$, $\psi(\mu,\tau)=(-\mu/(2\tau),-\mu^3/(2\tau)),h(y) = y^{-3/2}1(y>0)$,
>  * $T(y)=(y,1/y,\log y)$, $\psi(\mu,\tau)=(-\mu/(2\tau),-\mu^3/(2\tau),-3/2)$, $h(y)=1(y>0)$.
>
> If $h(y)$ is nonconstant, we can add an extra component in $T(y)$, and readout matrix $\Xi$ will have an additional column. We will revise the Sec. 4 accordingly.
>
> > **[R4Q4]** Checking Lem B.2, it appears exponential family has to satisfy a finite supremum property ...
>
> **[R4A4]** In Lem B.2, since $\theta$ is fixed and $p(\cdot \mid \theta)$ is continuous on a compact interval, the lemma does not need an extra boundedness assumption. However, for Thrm 4.1 to hold uniformly over contexts $Z^{(0)}$, we do need $\theta(Z^{(0)})$ to remain in a compact set $K$ and each $f_\theta$ be Lipschitz on $(a,b]$ uniformly over $\theta \in K$.
>
> > **[R4Q5]** What are the precise mathematical conditions ... for your results to hold?
>
> **[R4A5]** We appreciate the reviewer’s point and will make these assumptions for Thrm 4.1 explicit in the revision. We assume, uniformly over $Z^{(0)}$,
>
> 1. Solver: the setting of Thrm 3.1,
> 2. MLP.: $\theta(Z^{(0)})$ lies in a compact set $K$, and $\psi$ is approximated uniformly on $K$ by MLP $\tilde\psi$ with a non-polynomial activation,
> 3. Discr: the target PPD belongs to a 1D exp. family $f_\theta(y)$,
> 4. Trunc: equidistant grid, $f_\theta(y)$ is Lipschitz continuous on $(a,b]$.
>
> We do not require $\exp(\tilde\psi(\theta)^\top T(y))$ to be integrable wrt. $h$, since our construction uses a finite-dim. vec. followed by softmax.

---

> > ### Author Rebuttal · Reviewer_gA2c · 2026-04-04
> >
> > The authors fully resolved my minor concerns: thanks for this.
> >
> > Regarding my broader concern about existence style results common in the literature, I do not expect a satisfying answer in the rebuttal. I thank the authors for their continued efforts in highlighting the gap between existence and optimisation / learning.
> >
> > I will maintain my positive score.

---

> > > ### Author Response · Authors · 2026-04-06
> > >
> > > Thank you for your careful reading and for raising these important concerns. We appreciate the opportunity to clarify the scope and contributions of the paper, and your comments will help us improve the revision.

---

### Official Review · Reviewer_hVLx · 2026-03-09

**Soundness:** 2
**Presentation:** 3
**Significance:** 2
**Originality:** 2
**Overall Recommendation:** 4
**Confidence:** 3

**Summary:**

The paper provides a theoretical study of PFNs (Prior-fitted Networks; Müller et al., 2022) by showing by construction how transformers can learn the posterior predictive distribution. First, it shows that a sequence of attention layers can approximate the posterior predictive mean and variance. Then, from the latter moments, the model can approximate the full posterior predictive distribution. Based on the first step construction, the paper provides an explanation of why the original architecture struggles to generalize beyond the pretraining context sizes. The paper then proposes normalization of attention scores and depth scaling to mitigate the issue. The theoretical findings are complemented by empirical experiments, all conducted on a Gaussian Process prior with fixed hyperparameters (lengthscale, mean, variance) and input dimension.

**Compliance With Llm Reviewing Policy:**

Affirmed.

**Final Justification:**

My main concern about the limited validation of the findings was partially addressed during the rebuttal. The authors point to a possible extension of the contribution to other setups (hierarchical and latent GPs), which are arguably more interesting than the considered one. They also include an ablation on the lengthscale and discuss how it affects the interpretation. This is a particularly interesting insight to me.

My concern was partially addressed because I believe a non-negligible change is required to reflect these clarifications in the current paper. That said, I also think the contribution is solid, thus I revise my recommendation to acceptance. I hope the authors will incorporate these changes as they promised.

**Key Questions For Authors:**

- Could you explain how the theoretical results can be reused for a more realistic prior (e.g., a fully Bayesian GP or Bayesian neural networks)?
- Could you elaborate on the effect of lengthscale and variance on the empirical and theoretical results?
- Could you comment on how calibrated PFNs predictions are, and whether one can recover the same scaling behavior of calibration quality with respect to depth?

**Limitations:**

No, the paper does not discuss the limitations. The main limitation of the work is that both empirical and theoretical settings are limited.

**Strengths And Weaknesses:**

The main merit of the paper is its theoretical contribution and the resulting insights. Attention normalization is not a new idea, and multiple works have indeed explored normalization from transformers to improve stability and length generalization. The advantage of the paper, however, is that it characterizes the bottleneck and provides a theoretical argument for the specific case of a GP prior. While the considered setup is simplistic (see the comments below), I think the theoretical contribution of the paper still leads to a better understanding of Transformers. Additionally, the paper is well-structured and easy to follow. It appears that the experiments were conducted rigorously.

Despite the theoretical merits, I have a few concerns regarding the limited scope of the experiments, the evaluation metric, and novelty with respect to previous works.

- The results are constructed assuming Gaussian Process prior with fixed hyperparameters. I agree that this setup is perfect for small ablation study and controlled experiments. However, PFNs thrive when the prior is too complex, cases where traditional methods would not be tractable. This is the case for the main popular application of PFNs: TabPFN (Hollmann et al, 2022) using structural causal graph prior and PFNs4BO (Müller et al, 2023) using GP with a composite kernel or MLP priors.  The paper would benefit from a concrete discussion and illustration on how these theoretical  results could be applied to the more realistic setup. For example, would the same argument apply to fully Bayesian GP?
- Related to above point, even on a fixed hyperparameter GP prior, the experimental validation lacks a sensitivity analysis to assess whether the results can be reproduced with a varying setup. I think for instance the lengthscale, which intuitively controls the difficulty of the task, could be highly important. It seems that experimental validation (Section 6) could be extended to cover a broad setup.
- While the paper claims to focus on the approximation of the full posterior predictive distribution (last paragraph of Section 2), the experimental section only reports log MSE and 90% coverage. It appears to me that calibration error would be more appropriate in assessing the full PPD.  This is connected to how well the PFNs approximate variance, and if there is any expected behavior for the two extremes when the data is noiseless or highly noisy.
- The paper frames the characterization of PPD from moments and bin size scaling behavior as contributions (Section 4). It is however unclear to me what is the main novelty of this section. First, the paper only refers to the well-known universal approximation theorem to characterize the mapping from moments to PPD. Second, regarding the discretization, it is not surprising to me that increasing the number of bins leads to a better approximation of the continuous PPD. Maybe this section would benefit from a clear discussion about what the contribution is here.

Overall, I want to emphasize that the theoretical contribution of the paper, even though established on a restricted setup, is valuable for the community. However, it appears to me that further experiments or analysis are needed to understand how these results generalize to a real setup, where PFNs outperform standard approaches. In its current form, I recommend a rejection but I am happy to revisit the score if the above comments are resolved.

---

> ### Author Rebuttal · Authors · 2026-03-31
>
> We thank the reviewer for the thoughtful feedback. We address the main concerns below.
>
> > **[R3Q1]**
> > ... how the theoretical results ... for a more realistic prior ... would the same argument apply to fully Bayesian GP?
>
> **[R3A1]** We agree that our theory is limited to GP regression, and we will make this clearer. We focus on GPs because their PPD is available, letting us analyze approximation error, isolate architectural effects, and evaluate distributional metrics. GP regression is also a broad class, covering linear/nonparametric regression and Gaussian state-space models.
>
> We discuss two possible extensions of our construction to more complex priors, and will include this discussion in the revision.
>
> **1. Hierarchical GP.** Thrm 3.1 can be extended to fully Bayesian GP regressions with a fintie discrete prior on hyperparameters. For example, let $\theta \sim \sum_{h=1}^H \pi_h \delta_{\theta_h}$ and conditional on $\theta_h = (\ell_h,\sigma_h)$,
> $$
> f \sim GP(0,\kappa_{\ell_h}),y_i = f(x_i) + \epsilon_i,\epsilon_i\sim N(0,\sigma_h^2).
> $$
> This can be done by enlarging the token dimension and using multi-head attention, with each head performing componentwise recursion.
>
> |  | PPD | Token dim.  | # head  | output |
> |---|---|---|---|---|
> |GP   | $p_1(y\mid x, D_n)$   |  $d+4$ |  $1$ | $(\mu_1,\tau_1)$  |
> |Hierarchical GP   | $\sum_{h=1}^H w_hp_h(y\mid x, D_n)$  | $d+1+3H$  | $H$  | $(\mu_h,\tau_h)_{1:H}$  |
>
> $p_h(y \mid x, D_n)$ is the componentwise PPD, and $w_h$ denotes the mixture weight. We will provide the full derivation upon request. $w_h$ can, in principle, be achieved with an MLP. While we leave this extension to future work, our simulations in **[R1A3]** show that even the single-head construction performs promisingly in such setting.
>
>
> **2. Latent GP.** Please see **[R1A1]**.
>
> For more complex priors, our decomposition may still be useful, where attention computes PPD moments, and a shallow MLP maps them to bins. It can extend to richer priors if the relevant posterior quantities admit an iterative computation that attention can realize; e.g., in tree-based graphical model, marginal inference is often carried out by message passing, and prior work [3] has empirically shown that transformers can closely approximate such procedures.
>
> [3] Garnier-Brun et al. (2025). How transformers learn structured data: insights from hierarchical filtering. ICML
>
> > **[R3Q2]**
> > ... calibration error would be more appropriate ... how calibrated ... with respect to depth? / ... validation lacks a sensitivity analysis ... / ... the effect of lengthscale and variance ... ?
>
> **[R3A2]**
> We agree that the empirical section is limited, and we will include sensitivity analysis over $\ell,\sigma$ with more calibration metrics in the revision.
>
> The effects of lengthscale $\ell$ and variance $\sigma^2$ are both-sided. For $\ell$,
> * Smaller $\ell$ makes the GP rougher and localized, making prediction harder, especially in higher $d$. However, it makes the linear system easier to solve.
> * Larger $\ell$ makes the GP smoother, but can worsen conditioning due to correlations.
>
> We conducted additional simulations, fixing $\sigma=.2$ as in Sec. 6.3, $n_{\max}=256$, $d=16$, varying $\ell = .4,.8,1.6$ (including the paper setting $\ell=.8$ for comparison), and report more coverages and CRPS (continuous ranked probability score), a scoring rule for the predictive CDF (lower is better, absolute magnitude depends on scale). Entries are ordered by $L=8,16,32$.
>
> | $\ell$ | $n$ | CRPS | 50%  | 90%  | 95%  |
> |---:|---:|---|---|---|---|
> | .4 | 256  | .538, .547, .539 | .504, .487, .505 | .904, .899, .904 | .953, .946, .953 |
> |  | 1024 | .54, .54, .538 | .495, .485, .499 | .896, .887, .894 | .947, .943, .947 |
> | .8 | 256  | .318, .31, .307 | .488, .499, .509 | .89, .901, .905 | .942, .951, .954 |
> |  | 1024 | .271, .262, .258 | .444, .478, .505 | .848, .877, .897 | .908, .932, .95 |
> | 1.6 | 256  | .157, .151, .149 | .5, .498, .506 | .901, .9, .908 | .95, .948, .952 |
> |  | 1024 | .147, .137, .135 | .481, .479, .503 | .883, .884, .908 | .938, .943, .952 |
>
> In our setting, CRPS decreases as $\ell$ increases, indicating that larger $\ell$ reduces the statistical difficulty. Still, across all $\ell$, greater $L$ improves CRPS, most visibly at $\ell=.8$, while coverage remains reasonably close to nominal for $n>n_{\max}$. This result, together with **[R1A3]** under random hyperparameters, provides support for the proposed depth and normalized attention mechanisms.
>
> Larger $\sigma$ makes the prediction harder, but improves the conditioning via regularization. Additional runs at $\ell =.8$ with $\sigma=.1, .2, .4$ (https://anonymous.4open.science/r/tf-uq-rep-9DFA/tbl1.md) show that smaller $\sigma$ (sharper PPD) makes solver errors less forgiving, so shallow models under-cover more at larger $n$. Across all $\sigma$, CRPS improves with $L$.
>
>
> > **[R3Q3]**
> > ... as contributions (Sec 4). ... what the contribution is here.
>
> Please see **[R4A2]**.

---

> > ### Author Rebuttal · Reviewer_hVLx · 2026-04-03
> >
> > Thanks for your detailed response.
> >
> > The response addressed my concerns partially in the sense that it provides intuition, pointers, and small empirical validation. A full extension of the experiments to complex priors requires a major change indeed. For the revised version, I recommend that the authors add an elaboration discussion on the extent to which the setup can be applied. I will revise my score.

---

> > > ### Author Response · Authors · 2026-04-06
> > >
> > > Thank you very much for your thoughtful follow-up and for considering a score revision. We will add a discussion of potential extensions of our framework in the revision. We truly appreciate the time you have spent reading our paper and rebuttal. If you have a chance to update your score when convenient, we would be very grateful.

---

### Official Review · Reviewer_u4xx · 2026-03-13

**Soundness:** 3
**Presentation:** 3
**Significance:** 2
**Originality:** 3
**Overall Recommendation:** 5
**Confidence:** 2

**Summary:**

The authors show that a certain construction of transformers can implement a gradient descent algorithm targeting the posterior predictive mean and variance in Gaussian Process Regression problems. In this construction, the PPD is followed by nonlinear mappings that yield binned probabilities of the PPD distribution. Mathematical results concerning error bounds of the approximated PPD, the importance of normalization in self-attention, and the effect of depth on the distribution are studied. They conduct simulations explicitly testing and supporting their mathematical results.

**Compliance With Llm Reviewing Policy:**

Affirmed.

**Key Questions For Authors:**

1. Is there reason to believe your mathematics results apply to ICL problems beyond those with similar structure to Gaussian regression, e.g., linear regression, modeling finite state automata, or just arbitrary probabilistic models from which you can sample sequences?

2. Could you more explain why a researcher who works on transformers but not on PFNs specifically might be interested in these results?

3. What practical uses might this analysis be used for? For example, do your results suggest that transformer models trained on certain data could provide good estimates of posterior uncertainty?

**Limitations:**

Authors describe assumption of theorems but do not have clear limitations section. The paper does include impact statement.

**Strengths And Weaknesses:**

Strengths:
- The logic of the paper and its motivation is fairly clear
- Mathematical results are highly detailed with well organized and detailed appendices
- Plots are relatively easy to understand
- Experiments section directly tests all main theorems providing solid empirical support for the theoretical results
- The paper seems to add novel results concerning the relation between in-context learning (ICL) and transformers ability to estimate posterior predictive distribution


Weakness:
- It is not obvious how far the mathematical results generalize. In particular, the focus of analysis was on Gaussian process regression problems. It is not clear whether the analysis will be able to shed light on the way transformers perform ICL or estimate posterior predictive distributions beyond this narrow class of problems given Gaussian processes have an obvious formal similarity to self-attention. It is well-known, for example, that a self-attention attention layer can be described as performing a kind of Gaussian kernel regression or Gaussian process under certain assumptions (this even shows up in textbooks like Murphy's textbook "Probabilistic Machine Learning"). It is unclear whether the math results of the paper apply to data generating processes that do not have obvious formal similarities to self-attention.

---

> ### Author Rebuttal · Authors · 2026-03-31
>
> We thank the reviewer for the thoughtful feedback. We address the main concerns below.
>
> > **[R2Q1]**
> > It is not obvious how far the mathematical results generalize. In particular, the focus of analysis was on Gaussian process regression problems ...
> > Is there reason to believe your mathematics results apply to ICL problems beyond those with similar structure to Gaussian regression, e.g., linear regression, modeling finite state automata, or just arbitrary probabilistic models from which you can sample sequences?
>
> **[R2A1]** We agree that GP regression is an especially favorable setting, because its kernel structure makes the connection to self-attention transparent. We chose this setting deliberately: it allows us to make the posterior predictive computation explicit, decompose the approximation error, isolate architectural effects, and evaluate distributional metrics using the closed-form PPD.
>
> That said, we believe the contribution of our paper is nontrivial. We give an initial, constructive and quantitative analysis showing how transformer layers implement iterative posterior computation, how normalization enables context-length generalization, and how depth improves approximation quality. These conclusions are not straightforward from the GP-attention analogy alone.
>
> Moreover, GP is a broad class, encompassing linear and nonparametric regression, and Gaussian state-space models. Our construction can also be potentially extended to hierarchical GPs with random hyperparameters and latent GPs with non-Gaussian likelihoods; please see **[R3A1]** and **[R1A1]**. More broadly, the core of our framework is the decomposition where attention iteratively computes predictive quantities characterizing the PPD, and a shallow MLP maps them to bins. This suggests that related constructions may extend beyond GP settings whenever posterior prediction admits a structured sequential computation, such as message passing in graphical models.
>
> At the same time, we agree that the current paper does not establish a universal theory of ICL for arbitrary probabilistic models. We will make this limitation more explicit in the revision, and clarify that extensions to richer prior families are future directions, not claims of the current paper.
>
>
> > **[R2Q2]**
> > Could you more explain why a researcher who works on transformers but not on PFNs specifically might be interested in these results?
>
> **[R2A2]** Our results may interest transformer researchers beyond PFNs for studying transformer mechanisms for uncertainty-aware prediction. In the PFN setting targetting PPD with GP regression, the role of each component can be made explicit as in our construction: attention performs iterative computation of predictive quantities, the shallow MLP maps them to binned probabilities, and the overall pipeline yields a clear error decomposition. This, in turn, provides a mechanistic hypothesis for how transformers can realize uncertainty-aware prediction.
>
> The paper also studies architectural questions of broader interest. In particular, $n$-generalization that we study, where a PFN maintains distributional accuracy for contexts larger than pretraining sample range, is a form of context-length generalization that can be precisely measured via distributional metrics. Although our theorems are proved in the PFN setting, the roles of pretraining sample range, depth and attention normalization established here can potentially shed light on context-length generalization problem for transformers broadly.
>
>
>
> > **[R2Q3]**
> > What practical uses might this analysis be used for? For example, do your results suggest that transformer models trained on certain data could provide good estimates of posterior uncertainty?
>
> **[R2A3]** We see the main practical use of our analysis as offering design insight for PFNs. In practice, one must allocate limited model capacity across different architectural components. Our results suggest that, to the extent the construction extends to richer priors, increasing depth could be one principled way to improve distributional accuracy. While this is not entirely surprising, our contribution is to make this depth–accuracy connection explicit both theoretically and empirically.
>
> On the reviewer’s second point, we do not claim that our results guarantee PFNs are always reliable uncertainty estimators in practice. Rather, we provide support for this possibility in a controlled setting by giving an error decomposition across architectural components (Theorem 4.1) and validating it in simulation. In realistic settings, these guarantees may break down due to more complex priors or pretraining-inference distribution mismatch. Thus, our claim is more modest: in the controlled setting studied here, posterior uncertainty estimates can be made accurate because the transformer is capable of realizing the target computation, namely iterative computation of predictive moments followed by a discretized approximation of the PPD through MLP.

---

> > ### Author Rebuttal · Reviewer_u4xx · 2026-03-31
> >
> > The authors adequately answered my questions and concerns. I believe this work is rigorous and the authors made a significant contribution beyond previous work on PFNs and ICL (in the context of Gaussian Process Regression). I will raise my score.

---

> > > ### Author Response · Authors · 2026-04-03
> > >
> > > We are grateful to the reviewer for the time taken to engage with our work, and we are pleased that our responses resolved the reviewer's concerns.

---

### Official Review · Reviewer_VvBV · 2026-03-13

**Soundness:** 2
**Presentation:** 3
**Significance:** 2
**Originality:** 3
**Overall Recommendation:** 4
**Confidence:** 4

**Summary:**

The paper provides a theoretical analysis of how transformer-based PFNs can learn posterior predictive distributions in-context for Gaussian process regression. It shows by construction that attention layers can iteratively compute predictive mean and variance via a Richardson iteration scheme, and that a shallow output network can convert these quantities into a discretized predictive distribution. The paper further derives error bounds that clarify how approximation quality depends on attention depth and bin resolution, and studies how attention normalization (interpreted as Jacobi preconditioning) and depth affect generalization to larger context sizes than those seen during pretraining. These theoretical claims are supported by simulation results on Bayesian linear and RBF kernel regression tasks.

**Compliance With Llm Reviewing Policy:**

Affirmed.

**Final Justification:**

The paper provides a rigorous constructive proof that transformer-based PFNs can approximate GP posterior predictive distributions in-context. My main concerns were about the narrowness of scope (GP-specific theory) and limited empirical evaluation. The rebuttal addressed both: the hierarchical GP extension and latent GP discussion meaningfully broaden the theoretical reach, while the Sacramento dataset and hierarchical GP experiments strengthen the empirical case. The scope remains within the GP family, which limits significance somewhat, but the results are technically sound, clearly presented, and could open productive lines of inquiry into what PFNs are actually learning. I raise my score from 3 to 4.

**Key Questions For Authors:**

1. **How far do you believe the theoretical construction extends beyond Gaussian process regression?**
One of the interesting takeaways of the paper is that greater attention depth improves approximation quality and out-of-range generalization. At the same time, the iterative-solver interpretation suggests a potential scaling concern if increasingly large sample sizes require increasingly deep models due to conditioning of the Gram matrix. How do the authors see this tradeoff in relation to practical PFN deployments at larger n?

2. **How should the depth requirement be interpreted from a scalability perspective?**
   One of the interesting takeaways of the paper is that greater attention depth improves approximation quality and out-of-range generalization. At the same time, the iterative-solver interpretation suggests a possible scaling issue if increasingly large sample sizes require increasingly deep models due to conditioning of the Gram matrix. How do the authors see this tradeoff in relation to practical PFN deployments at larger \(n\)

3. **Do the authors have stronger evidence that the proposed mechanism is relevant beyond the synthetic BLR/RBF settings studied here?**  The experiments are well aligned with the theory, but the empirical validation is mostly based on controlled BLR and RBF setups, with Walker Lake as the only real-data case study. Do the authors have additional evidence, empirical or conceptual, that the same mechanisms around depth and normalization are operative in more complex non-Gaussian settings?

**Limitations:**

yes

**Strengths And Weaknesses:**

### Strengths

- The paper makes a genuine theoretical contribution. Rather than only arguing heuristically that PFNs may capture uncertainty in context, the authors give an explicit construction showing how transformer attention can implement an iterative solver for the GP posterior predictive mean and variance, followed by a shallow MLP that maps these moments to a discretized posterior predictive distribution. This is a meaningful extension of prior work that mainly focused on point prediction.

- The empirical section is well aligned with the theory. The authors directly test the quantities highlighted by the analysis, namely attention depth, bin resolution, and normalization for sample-size generalization, which makes the overall story coherent and well motivated.

- The normalization analysis is particularly interesting. Interpreting normalized attention as a form of preconditioning gives a concrete mechanistic explanation for why normalized models generalize better beyond the pretraining range.

- The paper is clearly written overall and easy to follow.

### Weaknesses

- The main limitation is scope. The theory is developed specifically for Gaussian process regression, which is a particularly favorable Bayesian setting because the posterior predictive distribution is analytically tractable. As a result, I do not think the paper should be interpreted as showing that transformers can generally perform Bayesian inference in context for more general prior families. Rather, the paper shows that transformers can approximate GP posterior predictive distributions in this specific setting, which is a narrower claim than a general result about Bayesian in-context learning.

- The depth result is interesting, but it also points to a scalability concern. If the required depth grows with sample size because convergence is limited by conditioning of the Gram matrix, then the iterative-solver interpretation may become less compelling in the larger-\(n\) regimes where PFNs are increasingly used in practice.

- The empirical evaluation supports the theory, but it still feels fairly limited in scope. The BLR and RBF experiments are sensible for validating the main claims, but they remain synthetic, and the single Walker Lake case study is not enough to demonstrate broad practical impact.

---

> ### Author Rebuttal · Authors · 2026-03-31
>
> We thank the reviewer for the thoughtful feedback. We address the main concerns below.
>
> > **[R1Q1]**
> > The main limitation is scope. ... **How far do you believe ... beyond GP regression?**
>
> **[R1A1]** We agree that our theory is limited to GP regression, and we will make this clearer. We focus on GP because their PPD is analytically tractable, which lets us analyze approximation error, isolate architectural effects, and evaluate distributional metrics in simulation. GP regression is also a broad class, covering Bayesian linear regression, nonparametric regression, and Gaussian state-space models.
>
> We discuss two possible extensions of our theoretical construction, and will include this discussion in the revision.
>
> **1. Hierarchical GP.** Theorem 3.1 can be extended to fully Bayesian GP regressions with a fintie discrete prior using multi-head attention. Please see **[R3A1]**.
>
> **2. Latent GP.** Our construction can also extend to latent GP regression, which includes GP classification, dynamic generalized linear models, and spatial/spatio-temporal models [1]. Concretely, suppose
> $$
> f \sim GP(0,\kappa_\theta),
> \quad
> y_i \mid f(x_i) \sim p(\,\cdot \mid f(x_i)),
> $$
> for a non-Gaussian $p(y\mid f)$. In many latent GP models, exact inference is intractable, but approximations yield $f(x)\mid D_n \approx N(\mu_x,\tau_x^2)$ that has direct connection to PPD, see Sec. 4 of [2]. Our construction applies to the approximate posterior of $f(x)$, and the idea of binning/softmax readout scheme that targets PPD could be similarly applied to certain PPDs, e.g. those obtained by thresholding latent GPs.
> * [1] Rue, H., Martino, S., & Chopin, N. (2009). Approximate Bayesian inference for latent Gaussian models by using integrated nested Laplace approximations. JRSS-B
> * [2] Chu, W., Ghahramani, Z.,(2005). Gaussian processes for ordinal regression. JMLR
>
>
> > **[R1Q2]**
> > The depth result is interesting, but .... **How should the depth requirement be interpreted from a scalability perspective?**
>
> **[R1A2]** We agree that Thrm 5.3 raises a scalability question. However, it gives only a *sufficient* condition in a deliberately constrained construction with a fixed geometry and a single step size. In contrast, our learnable transformer is substantially more flexible. At the same time, the $Q,K$ maps remain diagonal and the sparsity pattern is inherited from the theoretical construction, so the model can still be interpreted as Richardson-style iteration but with layerwise geometry, representations, and step sizes.
>
> Consistent with this view, Fig. 5 shows that generalization improves not only with $L$, but also with increasing $n_{\max}$ at a fixed $L$, since larger $n_{\max}$ exposes the learnable transformer to a wider range of task eigenspectra. We therefore view Thrm 5.3 as clarifying the role of $L$ in a simplified setting, not as claiming its scaling governs all practical PFNs.
>
>
> > **[R1Q3]**
> > The empirical evaluation supports the theory, but ... **Do the authors have stronger evidence that ... beyond the synthetic BLR/RBF settings studied here?**
>
> **[R1A3]** We agree that the empirical section in the current draft is limited in scope. At the same time, the RBF settings can be rich, covering the space of continuous functions on compact subsets of input space. That said, to provide further empirical evidence, we have conducted
> 1. additional evaluation in a fully Bayesian setting,
> 2. demonstration using Sacramento housing price data.
>
> We will include these results in the revision.
>
> **1. Additional evaluation.** We consider the hierarchical GP in **[R3A1]**. For each batch, we sample $\theta=(\ell, \sigma)\in \Theta$ from
>
> $$
> \Theta = \\{0.4, 0.8, 1.2\\} \times \\{0.1, 0.2, 0.3\\}, \quad \pi(\theta)=1/9,
> $$
>
> and generate the context. The PPD is tractable as a finite mixture of Gaussians. We used $n_{\max}=256$, $d=16$, otherwise the same as Sec 6.3. Entries are ordered by $L = 8,16,32$.
>
> | n | CRPS | 50% | 90% | 95% |
> |---:|---|---|---|---|
> | 256 | .38, .378, .377 | .573, .575, .59 | .895, .897, .901 | .937, .937, .942 |
> | 512 | .374, .371, .367 | .57, .568, .59 | .889, .891, .903 | .926, .925, .932 |
> | 1024 | .358, .353, .346 | .56, .563, .598 | .887, .887, .902 | .93, .929, .94 |
>
> The result is consistent with the paper: CRPS (continuous ranked probability score; lower CRPS is better, absolute value depends on scale) improves with $L$, while coverage remains reasonably calibrated for $n>n_{\max}$, especially at the 90% and 95% levels. This suggests that the depth effect and normalized-attention mechanism identified in the paper remain relevant in this more complex setting.
>
> **2. Additional data.** Sacramento dataset (?caret::Sacramento in R) is more challenging than Walker Lake because the spatial design is locally sparse. Despite this, the transformer PPD tracks that of exact GP closely overall except in the regions with no nearby observations. The figure is in https://anonymous.4open.science/r/tf-uq-rep-9DFA/fig1.pdf

---

> > ### Author Rebuttal · Reviewer_VvBV · 2026-04-03
> >
> > I thank the authors for the detailed rebuttal. My main concerns have been adequately addressed. I believe the results are interesting and could potentially open new avenues of research regarding what PFNs are actually learning. I raise my score from 3 to 4.

---

> > > ### Author Response · Authors · 2026-04-06
> > >
> > > Thank you again for your thoughtful questions and careful engagement with our rebuttal. We sincerely appreciate the time you have spent on our submission.

---

### Decision · Program_Chairs · 2026-04-30

**Decision:**

Accept (regular)

**Comment:**

The paper gives a constructive proof that transformer-based PFNs can approximate GP posterior predictive distributions in-context, and analyzes how depth, normalization, and output binning affect error and length generalization.

While the reviewers praise the explicit construction, the depth/normalization insights, and the experiments’ close alignment with the theory, they also raise several concerns, including that the theory is GP-specific and the empirical validation was initially mostly synthetic, leaving open how broadly the findings transfer to richer priors or more realistic PFN settings.

The authors' rebuttals added discussion of hierarchical and latent GP extensions, provided extra empirical evidence on Sacramento housing and hierarchical GP settings, added sensitivity analysis over lengthscale/noise, and clarified the existence-vs-learning scope of the theory.
The reviewers indicated that these responses have addressed most of their criticisms.

Therefore, we have decided to accept the paper for presentation at ICML. We would still recommend that the authors take the reviewers' feedback into account when preparing the camera-ready version.